METHODS AND RESOURCES

# Circadian regulation of the transcriptome in a complex polyploid crop

**Hannah Rees**[1], **Rachel Rusholme-Pilcher**[1], **Paul Bailey**[2], **Joshua Colmer**[1], **Benjamen White**[1], **Connor Reynolds**[1], **Sabrina Jaye Ward**[1], **Benedict Coombes**[1], **Calum A. Graham**[3,4], **Luíza Lane de Barros Dantas**[3], **Antony N. Dodd**[3], **Anthony Hall**[1] *

**1** Earlham Institute, Norwich Research Park, Norwich, United Kingdom, **2** Royal Botanic Gardens Kew, Richmond, Surrey, United Kingdom, **3** John Innes Centre, Norwich Research Park, Norwich, United Kingdom, **4** School of Biological Sciences, University of Bristol, Bristol, United Kingdom

* anthony.hall@earlham.ac.uk

**Data Availability Statement:** Data for the recreation of all main figures in this manuscript are available in S1 Data. Data for the recreation of all supplementary figures in this manuscript are

## Abstract

The circadian clock is a finely balanced timekeeping mechanism that coordinates programmes of gene expression. It is currently unknown how the clock regulates expression of homoeologous genes in polyploids. Here, we generate a high-resolution time-course dataset to investigate the circadian balance between sets of 3 homoeologous genes (triads) from hexaploid bread wheat. We find a large proportion of circadian triads exhibit imbalanced rhythmic expression patterns, with no specific subgenome favoured. In wheat, period lengths of rhythmic transcripts are found to be longer and have a higher level of variance than in other plant species. Expression of transcripts associated with circadian controlled biological processes is largely conserved between wheat and *Arabidopsis*; however, striking differences are seen in agriculturally critical processes such as starch metabolism. Together, this work highlights the ongoing selection for balance versus diversification in circadian homoeologs and identifies clock-controlled pathways that might provide important targets for future wheat breeding.

## Introduction

Circadian clock homologs have been both inadvertently selected during crop domestication and identified as crop improvement targets [1–4]. Understanding circadian regulation of the transcriptome in crops such as bread wheat (*Triticum aestivum*) may provide useful insights for future crop improvement. Wheat also provides an excellent model system to explore how the circadian clock and its outputs are coordinated in a recently formed, complex allopolyploid. In *Arabidopsis*, circadian transcription factors act in a dose-dependent manner, with both knock-out and overexpression mutants resulting in altered function of the circadian oscillator [5–8]. It is not yet understood how rhythmic gene expression is balanced in species with multiple copies of the same gene. *T. aestivum* is a hexaploid (AABBDD) formed through interspecific hybridisation of 3 diploid ancestors around 10,000 years ago [9,10]. Approximately 51.7% of high-confidence wheat genes still exist in triads; sets of 3 homoeologous genes present on each of the A, B, and D genomes [11]. As these homoeologs evolved independently

available in S2 Data. Fastq data from the RNA-seq circadian time course are available to view from the Grassroots Data Repository: https://opendata. earlham.ac.uk/opendata/data/wheat_circadian_ Rees_2021. A summary csv table with expression of wheat genes (TPM), Metacycle estimates, gene annotations and triad balance classification can be viewed in S11 Table is available here: https:// opendata.earlham.ac.uk/opendata/data/wheat_ circadian_Rees_2021. A TPM expression matrix at the individual replicate level with triad IDs information is also available: S12 Table: https:// opendata.earlham.ac.uk/opendata/data/wheat_ circadian_Rees_2021. Code for creating Loom plots, the cross-correlation analysis and the clustering analysis are available from our groups GitHub repository: https://github.com/AHallLab/ circadian_transcriptome_regulation_paper_2022/ tree/main.

**Funding:** H.R., R.R.P. and A.H. were funded by the BBSRC Core Strategic Programme Grant (Genomes to Food Security) BB/CSP1720/1 and its constituent work package, BBS/E/T/000PR9819 (WP2 Regulatory interactions and Complex Phenotypes). B.W., R.R.P. and A.H. was supported by the BBSRC Designing Future Wheat grant BB/ P016855/1; BBS/E/T/000PR9783 (DFW WP4 Data Access and Analysis). B.C and J.C were supported by the BBSRC funded Norwich Research Park Biosciences Doctoral Training Partnership grant BB/M011216/1. C.R. by a BBSRC grant BB/ V509267/1 and Wave 1 of The UKRI Strategic Priorities Fund under the EPSRC Grant EP/ T001569/1, particularly the "AI for Science" theme within that grant & The Alan Turing Institute. We would also like to acknowledge BBS/E/T/ 000PR9816 (NC1 - Supporting EI's ISPs and the UK Community with Genomics and Single Cell Analysis) for data generation and BB/CCG1720/1 for the physical HPC infrastructure and data centre delivered via the NBI Computing infrastructure for Science (CiS) group. P.B. was supported by a BBSRC TRDF grant BB/N023145/1. A.N.D., L.L.B. D. and C.A.G. are funded by BBSRC ISP Genes in the Environment (BB/P013511/1). C.A.G. was also funded by UK BBSRC SWBIO DTP (BB/M009122/ 1). The funders had no role in study design, data collection and analysis, decision to publish, or preparation of the manuscript.

**Competing interests:** The authors have declared that no competing interests exist.

**Abbreviations:** AGPase, ADP-glucose pyrophosphorylase; BH, Benjamini–Hochberg; CT, circadian time; HMM, hidden Markov model; iToL, Interactive Tree of Life; PEP, plastid-encoded RNA polymerase; PS, photosystem; RAE, relative

for several million years prior to hybridisation, it is plausible that these independent species might have been subject to different selective pressures on their clocks (Fig 1A).

The circadian network in *Arabidopsis* comprises a series of interlocking negative transcriptional feedback loops connected by key activators [12]. Although monocots such as wheat diverged from their dicot relatives over 140 million years ago [13], many circadian oscillator components seem to have been conserved, particularly those forming the core loop network. Orthologs of *TIMING OF CAB EXPRESSION 1 (TOC1)* and other *PSEUDO-RESPONSE REGULATOR (PRR)* genes have been identified in wheat, rice, and barley, and several loci within these genes have been associated with altered flowering times, most notably the *ppd-1* locus within *TaPRR3/7* [14–16]. Likewise, mutants of orthologs of *LATE ELONGATED HYPOCOTYL (LHY)*, *GIGANTEA (GI)*, *EARLY FLOWERING 3 (ELF3)*, and *LUX ARRYTHMO (LUX)* have been identified that alter heading dates, pathogen susceptibility, plant height, or lower grain yields [17–21].

Circadian control of carbon fixation and starch metabolism are thought to form part of the selective advantage conferred to *Arabidopsis* by the clock [22,23]. This is apparent in the *lhy⁻/ cca1⁻* short period double mutant in *Arabidopsis*, where night-time starch levels reach exhaustion earlier compared to wild type, triggering early onset starvation responses that reduce plant productivity [23]. Similarly, genes encoding photosynthesis-related proteins are well-established targets of the circadian clock and include the *LIGHT HARVESTING CHLOROPHYLL A/B BINDING PROTEIN* genes (*LHCB* also known as *CAB* genes) and photosystem I and II reaction centre genes [24,25].

Here, we investigate circadian balance within wheat triads to understand how circadian control is coordinated in a polyploid crop with 3 subgenomes. Second, we examine similarities and differences between the circadian transcriptome in wheat and its distant dicot relative *Arabidopsis*, at a global level and at the level of genes encoding key pathways such as primary metabolism and photosynthesis.

## Results

### Experimental context

We generated a 3-day circadian RNA-seq time course, sampling all aerial tissue from wheat seedlings cv. Cadenza, using 3 biological replicates, every 4 h, following transfer to constant light conditions (CT0-68h). Transcripts were quantified at the gene level and were averaged across biological replicates. The entire time series (from 0 to 68 h) provides information about how gene expression changes upon transfer to constant conditions and is used in our triad analysis to identify differential patterns of circadian expression between homoeologs within wheat triads. A shorter part of the dataset (24 to 68 h) is used for calculation of circadian waveform characteristics to ensure appropriate quantification of rhythms is under free-running conditions and is also used in all cross-species comparisons.

### Global analysis of the circadian transcriptome in wheat

We compared the proportions of rhythmic genes oscillating under free-running conditions (CT24-68h) with a recently published dataset from *Arabidopsis* [26] (Table 1). Rhythmicity was assessed using Metacycle Benjamini–Hochberg (BH) $q$-values [27]. Of the 86,567 genes expressed in wheat, 33.0% were rhythmically expressed with a BH $q < 0.05$ and 21.5% with a BH $q < 0.01$. This was significantly lower than the proportions of rhythmically expressed genes in the *Arabidopsis* dataset (50.7% BH $q < 0.05$, 39.1% BH $q < 0.01$) using the same criteria ($X^2$ [1] = 2,727.1, $p < 0.001$, 1-tailed, 2-proportions z-test). The proportions of rhythmic genes were higher following a stricter classification of gene expression (S1 Note and S1 Table);

amplitude error; RCA, Rubisco activase; SS, starch synthases; TF, transcription factor; TFBS, transcription factor binding site; TSS, transcription start site.

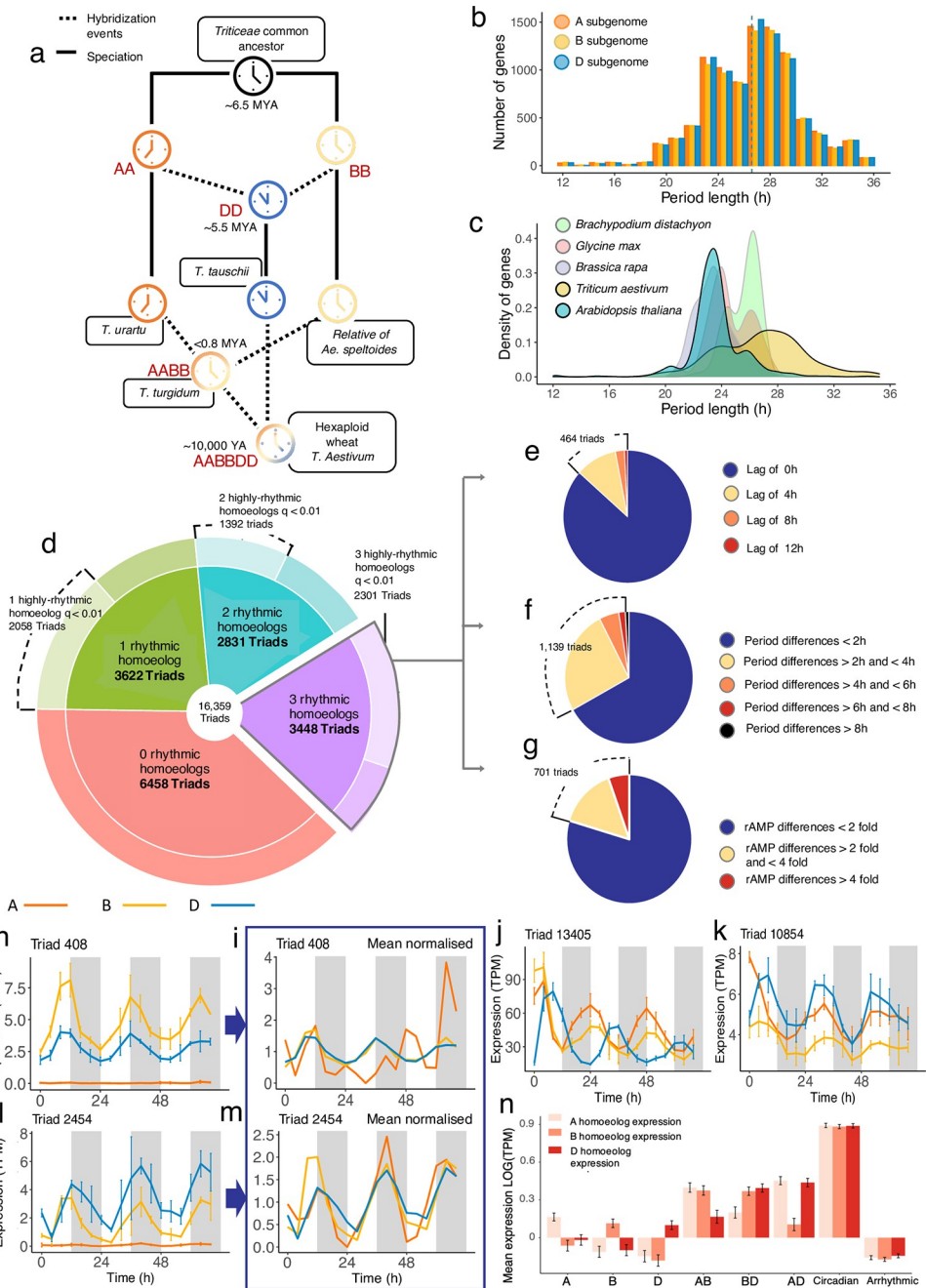

**Fig 1. Circadian regulation of homoeolog expression of wheat triads. (a)** Circadian clocks evolved independently in the ancestors of hexaploid wheat following divergence from a common ancestor approximately 6.5 million years ago. Colours of clock icons represent theoretical differences in clock regulation integrated in the tetraploid and hexaploid hybrids either through circadian balance or through dominance of a particular homoeolog copy. Speciation and hybridisation event dates are based on estimates from [115]. **(b)** Histogram showing distribution of period lengths in wheat split between the A, B, and D subgenomes. Periods were measured from 24–68 h data. Dotted line indicates the mean period for the A, B, and D subgenomes. (Data_Fig 1b in S1 Data). **(c)** Density plot showing the distribution of period lengths across rhythmic transcripts (BH $q < 0.01$) in *Arabidopsis*, *Brassica rapa*, *Brachypodium distachyon*, *Glycine max* (soybean), and wheat based on meta2d estimates on 24–68 h data following transfer to constant light. (Data_Fig 1c in S1 Data). **(d)** Proportions of triads with either 0 (red segment), 1 (green segment), 2 (blue segment), or 3 (purple segment) rhythmic gene(s) out of the 16,359 expressed triads in this dataset. Lighter shading in the outer segments represents cases where 1/2 homeolog(s) have high confidence rhythmicity (BH $q < 0.01$) alongside an arrhythmic homoeolog (BH $q > 0.05$). We term these genes "imbalanced rhythmicity" triads. Triad differences are based on meta2d estimates from data 0–68 h after transfer to L:L. Of the 3,448 triads with 3 rhythmic genes

(represented by the purple segment in (d)), we also looked for triads with circadian imbalance in: phase (**e**), period (**f**), or relative amplitude (**g**). A total of 464 triads had homoeologs which peaked with an optimum lag of 4, 8, or 12 h following cross-correlation analysis. A total of 1,139 triads had homoeologs with period differences of more than 2 h, and 701 triads had homoeologs with a more than 2-fold difference in relative amplitude. (**h**) An example of a triad with imbalanced rhythmicity, where 2 homoeologs are rhythmic and 1 is arrhythmic as can be seen when the homoeologs are mean normalised (**i**). (**j**) Example triad where the D-genome homoeolog lags by 8 h. (**k**) Example triad where the A genome homoeolog has a period estimate 4 h longer than the D-genome homoeolog. (**l**) Example triad where the relative amplitude of the D-genome homoeolog is more than 4 times that of the A-genome homoeolog, but the A-genome homoeolog is still rhythmically expressed (**m**). In plots h, j, k, and l, error bars on each plot represent standard deviation between 3 biological replicates. (Data_Fig 1h-m in S1 Data). Genes in example triads are: [Triad 408: TraesCS3A02G533700, TraesCS3B02G610500, TraesCS3D02G539000], [Triad 13405: TraesCS6A02G269100, TraesCS6B02G296400, TraesCS6D02G245800], [Triad 10854: TraesCS6A02G166500, TraesCS6B02G194000, TraesCS6D02G155100], and [Triad 2454: TraesCS2A02G333000, TraesCS2B02G348800, TraesCS2D02G329900]. (**n**) Mean expression of transcripts across all time points in the A, B, and D subgenomes within imbalanced rhythmicity triads compared with circadian balanced and arrhythmic triads. Error bars represent standard error. (Data_Fig 1n in S1 Data).

however, the proportions of rhythmic genes in wheat remained significantly lower than in *Arabidopsis*, regardless of the cutoff.

Circadian waveform characteristics of the rhythmically expressed genes (BH $q < 0.01$) in the wheat and *Arabidopsis* datasets were quantified using 4 algorithms in Metacycle [27] (JTK, ARSER, LS, and meta2d) and 2 algorithms in Biodare2 [28] (FFT-NLLS and MESA). Period, phase, and amplitude estimates from FFT-NLLS and meta2d were well correlated for individual genes (S1 Fig). All models reported that mean period length in wheat was approximately 3 h longer than in *Arabidopsis* (wheat = 25.9 to 27.5 h, *Arabidopsis* = 22.6 to 24.4 h; t(36067) = 101.58, $p < 0.001$, Welch's 2 sample $t$ test; S2 Fig). There was no significant difference between mean periods across the 3 wheat subgenomes (Fig 1B, F(2, 28,276) = 0.179, $p = 0.836$, 1-way ANOVA).

Periods, relative amplitudes, and $q$-values are estimates from meta2d. Data windows reflect hours relative to transfer to constant light from entrained 12:12h light conditions. A repeat of this table with pre-filtering to remove low-expression genes is provided in S1 Table, and the effects on proportions of rhythmic genes are discussed in S1 Note.

We wished to determine whether the longer period within the transcriptome was specific to wheat compared to other previously published plant circadian RNA-datasets. We found that period lengths in wheat were longer and had higher standard deviation than period distributions from *Arabidopsis*, *Brassica rapa*, *Glycine max*, and even the closely related diploid wild grass, *Brachypodium distachyon* (Fig 1C and S2 Table).

**Table 1. Numbers of rhythmic genes at (BH $q < 0.05$ or BH $q < 0.01$) in *Arabidopsis* and wheat identified using Metacycle BH $q$-values.**

| | Wheat data from this study | | *Arabidopsis* data from Romanowski and colleagues (2020) |
|---|---|---|---|
| | **24–68 Data window** | **0–68 Data window** | **24–68 Data window** |
| Total number of expressed genes | 86,567 | 86,567 | 26,392 |
| Total rhythmic genes (BH $q < 0.05$) | 28,594 | 28,530 | 13,392 |
| Total rhythmic genes (BH $q < 0.01$) | 18,633 | 21,059 | 10,317 |
| Mean period (h) (BH $q < 0.05$) | 26.60 h (SD 3.62) | 26.75 h (SD 2.82) | 23.50 (SD 2.52) |
| Mean period (h) (BH $q < 0.01$) | 26.82 h (SD 3.21) | 26.83 h (SD 2.42) | 23.62 (SD 2.04) |
| Mean relative amplitude (BH $q < 0.05$) | 0.24 (SD 0.19) | 0.26 (SD 0.20) | 0.28 (SD 0.20) |
| Mean relative amplitude (BH $q < 0.01$) | 0.27 (SD 0.19) | 0.29 (SD 0.21) | 0.30 (SD 0.20) |

We next investigated how mean wheat periods in the rhythmic transcriptome changed over the 3-day experiment using a 44-h sliding window and found that periods were longer immediately after transfer to constant light (0 to 44 h, 28.61 h, SD = 3.421 h) and progressively shortened over the following days (S2 Note).

For both *Arabidopsis* and wheat, we recalculated phases of rhythmic transcripts relative to endogenous period (circadian time; CT). Across all algorithms, most transcripts in *Arabidopsis* peaked during the subjective night (around CT12-24; S3 Fig). In wheat, the greatest numbers of rhythmic genes peaked during the subjective day (around CT6-8) with a second, smaller group being expressed in the night (~CT20). When we grouped transcripts into 2-h period bins, we found that transcripts with short periods contained proportionally more dawn-peaking transcripts, whereas those with longer periods contained proportionally more dusk-peaking transcripts (S3 Note and S4 Fig). These data suggest that dusk may be an important signalling cue for wheat circadian expression. It is possible that the mean period of dusk expressed transcripts initially lengthens trying to follow this missing dusk signal before the free-running endogenous period asserts itself. This may be one explanation as to why the mean period length starts off long and shortens over time in constant light and why transcripts with the longest period lengths tend to have dusk peaking expression.

## Balance of circadian regulation within triads

Although previous studies have examined the relationships between circadian regulated orthologs in different plant species [26,29,30] and within paralogs in *Brassica rapa* [31], hexaploid wheat provides an opportunity to study the relationships between recently formed circadian regulated homoeologs acting within the same organism. In wheat, over 72% of syntenic triads are estimated to have "balanced" expression, with similar relative abundance of transcripts from each of the 3 homoeologs [32]. Due to the importance of the clock in coordinating dosage of gene expression, our hypothesis was that many circadian triads would also have balanced circadian regulation. We defined imbalanced circadian regulation as triads harbouring differences in rhythmicity (i.e., BH $q$-values), period lengths, phases, and relative amplitudes across the full dataset from CT0-68h.

Of the 16,359 expressed triads in our dataset, 9,901 (60.52%) had at least 1 rhythmic homoeolog and 3,448 (21.08%) had 3 rhythmically expressed genes (BH $q < 0.05$), with the latter hereafter termed "rhythmic triads" (purple segment, Fig 1D). A total of 6,453 triads lacked rhythmicity in either 1 or 2 expressed homoeolog(s) (green and blue segments, Fig 1D). For triads where 1 gene is arrhythmic, there was no significant bias for absence of rhythmicity in the A, B, or D copy ($\chi2[2]$ = 1.8501, $p$ = 0.40). For triads where 2 genes are arrhythmic, there were slightly more triads where the rhythmic gene was in the A or B subgenome (755 or 749 triads, respectively) compared to the D subgenome (666 triads) $\chi2[2]$ = 6.8415, $p$ = 0.03. We found cases where high-confidence rhythmic homoeologs (BH $q < 0.01$) occurred alongside arrhythmic homoeologs (BH $q > 0.05$) represented by light-shaded outer-ring segments in Fig 1D. In total, there were 3,450 of these imbalanced-rhythmicity triads (Fig 1H and 1I). To explore other forms of circadian imbalance, we assessed whether phase, period, and relative amplitude were conserved between homoeologs within the rhythmic triad set (purple segment, Fig 1D). Differences in phases were quantified by a cross-correlation analysis to assess whether the correlation between homoeologs was improved with a time lag of 4, 8, or 12 h. We identified 464 triads with imbalanced phases with an optimum lag of >0 h between homoeologs (Fig 1E and 1J). A total of 1,139 triads had imbalanced periods with more than 2 h difference in period between homoeologs (Fig 1F and 1K), and 701 triads had imbalanced relative amplitudes with more than 2-fold difference in relative amplitude (Fig 1G and 1L). Within this last

group, the homoeolog with the lowest amplitude was still rhythmic, as observed when data are mean-normalised (Fig 1M). In summary, the largest cause of imbalanced circadian expression within triads was absence of rhythmicity (67.89%) with differences in period (22.41%), relative amplitude (13.79%), and phase (9.13%) occurring more infrequently and with some overlap between categories.

Out of all expressed triads in our dataset, around 11.1% had balanced circadian expression, 31.1% had imbalanced circadian expression, 39.5% were arrhythmic, and 18.4% were borderline triads that did not fit into the categories imposed by our cutoffs. There is therefore a ratio of approximately 3:1 imbalanced to balanced circadian triads in wheat. This finding was initially surprising given that Ramírez-González and colleagues [32] reported 72.5% of wheat triads showed balanced expression. We found that 64.15% of the triads classified as circadian imbalanced in our data would be classified as balanced in the Ramírez-González and colleagues [32] study (S5 Fig). However, if we consider that triads with highly imbalanced circadian regulation can be classified as balanced in their expression at a single time point (as demonstrated in S6 Fig) and that there are multiple ways in which homeologs can become circadian imbalanced (phase, period, rhythmicity, etc.), then it is quite reasonable that only a small proportion of triads are classified as having balanced circadian regulation in this study. This insight highlights the importance of considering temporal dynamics when studying gene expression.

One explanation for imbalanced rhythmicity is that arrhythmic homoeolog(s) are silenced. In support of this, we found that the rhythmic homoeologs in imbalanced triads were expressed at a significantly higher baseline level than their arrhythmic homoeologs (Fig 1N; $F(16, 35,148) = 6.94$, $p < 0.001$, 2-level, nested ANOVA on Log10 transformed data). We also found that triads with balanced rhythmicity were expressed at a uniformly higher level than the most highly expressed homoeolog(s) in the imbalanced triads (Fig 1N; $F(7, 35,148) = 570.909$, $p < 0.001$). Therefore, in imbalanced rhythmicity triads, the rhythmic homoeolog does not appear to compensate for reduced expression of the other homoeolog(s), so the overall expression across the triad is reduced. This is supported by data from diploid *Brassica rapa*, where circadian regulated paralogs are expressed at a higher level than single copy genes [31].

To investigate whether certain biological processes were associated with circadian balance, we compared GO-slim terms enriched in the 1,816 circadian balanced triads, the 5,082 differently circadian regulated triads, and the 6,458 arrhythmic triads, identifying significant terms unique to each group (Table 2). Some terms were enriched only in circadian balanced triads (*p*-value <0.0001, Fisher's exact test, e.g., "photosynthesis," "generation of precursor metabolites and energy," "gene-expression," and "translation"). In contrast, GO-slim terms: "developmental process involved in reproduction," and "system development" were enriched significantly in triads with differently regulated homoeologs (*p*-value <0.0001) but not in triads with circadian balanced homoeologs (*p*-value >0.5). A possible explanation for this enrichment could be that imbalanced circadian triads are more likely to be dynamically expressed over developmental stages or show local dominance of a subgenome in a particular tissue type. Transcription factor (TF) triads were as likely to be circadian balanced/imbalanced as non-transcription factors ($\chi^2 (2, N = 13,356) = 3.03$, $p = 0.08$, chi-square test). Previously validated wheat TFs with imbalanced circadian expression included *WCBF2* (aka *TaCBF1*) and *TaPCF5*, both of which regulate abiotic stress responses [33,34] (S4 Note and S7 Fig).

When we compared GO-slim terms associated with paralogs in *Brassica rapa* [31], we found that *Brassica* paralogs with similar circadian expression patterns (as characterised by Greenham and colleagues [31] using DiPALM) were also associated with "photosynthesis" and "generation of precursor metabolites and energy" (*p*-value <0.001, Fisher's exact test). These terms were not enriched in circadian paralogs with differential expression patterns (*p*-value

**Table 2. GO-slim terms for biological processes associated with circadian balanced, circadian imbalanced, and arrhythmic wheat triads.**

| | GO ID | Terms | p-Value in circadian balanced triads | p-Value in circadian imbalanced triads | p-Value in non-rhythmic triads |
|---|---|---|---|---|---|
| CIRCADIAN BALANCED | GO:0009628 | Response to abiotic stimulus | 0.00 | 0.22 | 0.58 |
| | GO:0015979 | Photosynthesis | 0.00 | 1.00 | 1.00 |
| | GO:0006091 | Generation of precursor metabolites and energy | 0.00 | 1.00 | 1.00 |
| | GO:0006518 | Peptide metabolic process | 0.00 | 1.00 | 0.92 |
| | GO:1901566 | Organonitrogen compound biosynthetic process | 0.00 | 1.00 | 0.92 |
| | GO:0009059 | Macromolecule biosynthetic process | 0.00 | 1.00 | 0.99 |
| | GO:0006412 | Translation | 0.00 | 1.00 | 0.98 |
| | GO:0034645 | Cellular macromolecule biosynthetic process | 0.00 | 1.00 | 1.00 |
| | GO:0010467 | Gene expression | 0.00 | 1.00 | 0.90 |
| | GO:0019725 | Cellular homeostasis | 0.00 | 0.54 | 0.96 |
| | GO:0065008 | Regulation of biological quality | 0.00 | 0.71 | 1.00 |
| CIRCADIAN IMBALANCED | GO:0003006 | Developmental process involved in reproduction | 0.85 | 0.00 | 0.87 |
| | GO:0090567 | Reproductive shoot system development | 0.92 | 0.01 | 1.00 |
| | GO:0009719 | Response to endogenous stimulus | 0.97 | 0.01 | 0.10 |
| | GO:0048731 | System development | 0.98 | 0.00 | 0.97 |
| | GO:0048608 | Reproductive structure development | 0.98 | 0.00 | 0.97 |
| | GO:0043412 | Macromolecule modification | 1.00 | 0.00 | 0.28 |
| | GO:0022414 | Reproductive process | 1.00 | 0.00 | 0.47 |
| NON-RHYTHMIC | GO:0044237 | Cellular metabolic process | 0.08 | 1.00 | 0.00 |
| | GO:0009605 | Response to external stimulus | 0.41 | 0.39 | 0.00 |
| | GO:0009607 | Response to biotic stimulus | 0.58 | 0.60 | 0.01 |
| | GO:0060255 | Regulation of macromolecule metabolic process | 0.60 | 0.76 | 0.00 |
| | GO:0009056 | Catabolic process | 0.65 | 0.93 | 0.00 |
| | GO:0048869 | Cellular developmental process | 0.71 | 0.19 | 0.00 |
| | GO:0019222 | Regulation of metabolic process | 0.78 | 0.85 | 0.00 |
| | GO:0044238 | Primary metabolic process | 0.81 | 0.24 | 0.00 |
| | GO:0071704 | Organic substance metabolic process | 0.81 | 0.26 | 0.00 |
| | GO:0008219 | Cell death | 0.82 | 0.94 | 0.00 |
| | GO:0010468 | Regulation of gene expression | 0.88 | 0.15 | 0.00 |
| | GO:0009790 | Embryo development | 0.95 | 0.98 | 0.00 |
| | GO:0007049 | Cell cycle | 0.97 | 1.00 | 0.00 |
| | GO:0065009 | Regulation of molecular function | 1.00 | 1.00 | 0.00 |
| | GO:0006807 | Nitrogen compound metabolic process | 1.00 | 1.00 | 0.00 |

Only enriched terms which were highly enriched (Fisher's exact test $p < 0.01$) in 1 category and nonsignificantly expressed ($p > 0.05$) in other categories is displayed.

>0.5; S3 Table). Examples of genes having similar (balanced) circadian expression within *Brassica rapa* paralogs and in all 3 homoeologs in wheat triads included orthologs of the PSI light harvesting complex genes *LHCA1*, *LHCA2* and *LHCA3*, and the RNA polymerase sigma factor *SIG5*. It is possible that conservation of circadian expression of these duplicate genes poses an

evolutionary benefit, as similar regulation has been retained for both duplicate copies in ancient paralogs of *Brassica rapa* and within more recently formed wheat homoeologs arising through polyploidization.

## Patterns of triad circadian balance across the genome

Next, we wanted to determine whether certain genomic regions incorporate a physical clustering of circadian balance in rhythmicity. We hypothesised that if blocks of sequential triads with imbalanced rhythmicity are present within particular chromosomal regions, this might indicate differential chromatin accessibility or transcriptional suppression. To investigate this, we identified regions on each set of chromosomes where there were sequential triads having the same number of rhythmic homeologs (i.e., runs of 1, 2, or 3 rhythmic gene(s)). We also looked for runs of sequential rhythmic triads specific to a particular chromosome (i.e., runs of 1, 2, or 3 rhythmic genes specifically on chromosomes A, B, or D). In both cases, we found no evidence that triads with specific numbers of rhythmic homoeologs were grouped together more often than would be expected by chance (S5 Note and S4 Table). This suggests that distributions of rhythmic balance appear to be randomly distributed across the genome (S8 Fig).

## Clustering of gene expression and GO-term enrichment

To establish whether similar phased transcripts in wheat and *Arabidopsis* had similar biological roles, we carried out a co-expression analysis to cluster rhythmic transcripts (BH $q < 0.01$). Unsupervised clustering using WGCNA [35], identified 9 expression modules for each species and we identified GO-slim terms enriched in each module ($p < 0.01$). Circadian characteristics of module eigengenes are shown in S5 Table. We compared the correlation and cross-correlation of pairwise modules in the 2 species to find modules that correlated with a peak lag of 0 (synchronous phase) or with a peak lag of 4, 8, or 12 h (asynchronous phase). Overall, modules with synchronous phases in wheat and *Arabidopsis* shared more GO-slim terms than modules with asynchronous phases (F(3,77) = 4.79, $p < 0.01$, 1-way ANOVA), indicating that these modules in wheat and *Arabidopsis* contain genes with similar functions (S9 Fig). We focused on 4 of these synchronous module pairs, broadly peaking at dawn, midday, dusk, and night for further analysis (Fig 2). Eigengenes for dawn peaking modules A9 and W9 were highly correlated (r > 0.9) and shared 14 overlapping enriched GO-slim terms ($p < 0.01$; orange colours in Fig 2A, 2B and 2F). These included terms for translation and gene expression as well as terms related to protein, amide, nitrogen, and organonitrogen biosynthetic and metabolic processes (full lists in S6 Table). Co-expressed genes in the dawn-expressed modules included several orthologs involved in light, heat, and biological defence, as well as 45 ribosomal protein orthologs (S6 Note). Transcripts for ribosomal proteins in mouse liver and *Neurospora crassa* have also been reported to oscillate, suggesting a conserved role for the circadian clock in coordinating ribosome biogenesis [36,37]. In addition to enriched GO-slim terms, we investigated enrichment for TF superfamilies and transcription factor binding site (TFBS) superfamilies in each wheat module of clock-regulated genes. In late-night/dawn modules W8 and W9, transcripts encoding MYB TFs were significantly enriched and included putative TFs involved in leaf morphogenesis, plant growth, regulation of flavonoid biosynthesis, and developmental transition to flowering (S6 Note and S10 Fig).

Eigengenes for wheat and *Arabidopsis* modules peaking in the day (W3 and A2) had a relatively low correlation (*r* = 0.491), but peaked with similar CT values (CT 6.34 h and 6.19 h) given the longer circadian period in wheat, and 5 out of 15 of the GO-slim terms enriched in the W3 module were also found in the A2 module ($p < 0.01$; yellow colours in Fig 2A, 2C and 2F). These included terms relating to "photosynthesis," "response to radiation," and

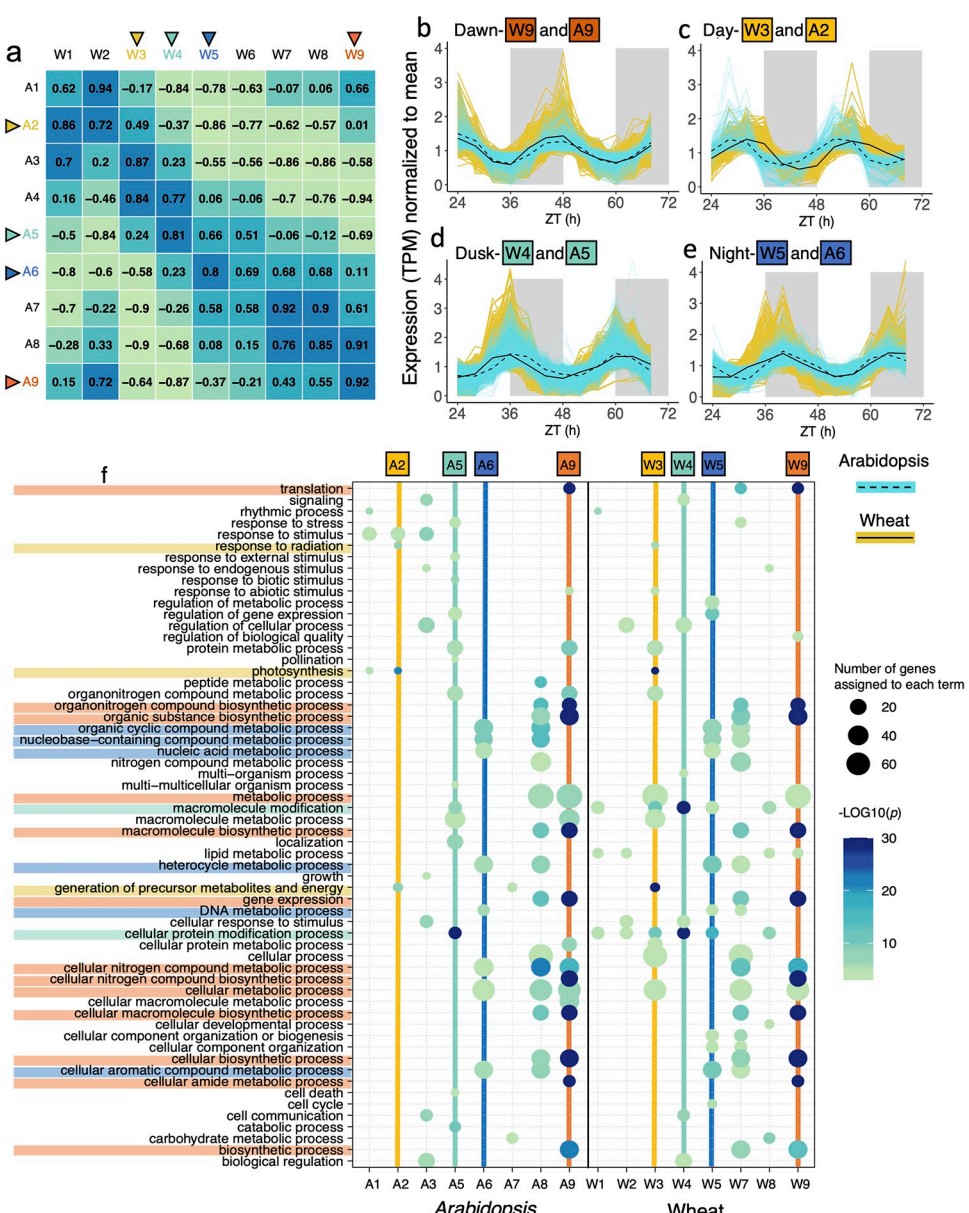

**Fig 2. Overlapping GO-slim terms shared between *Arabidopsis* and wheat modules expressed at similar times in the day.** Selected pairs of modules discussed in the main text in *Arabidopsis* and wheat are expressed at dawn (orange), day (yellow), dusk (turquoise), and at night (dark-blue). (a) Pearson correlation coefficient (r) between eigengenes for wheat and *Arabidopsis* expression modules ordered by circadian phase. (b–e) Expression profiles of selected *Arabidopsis* and wheat modules normalised to their mean. Solid and dashed black lines represent the module eigengene for wheat and *Arabidopsis* modules, respectively. (Data_Fig 2b-e in S1 Data). (f) GO-slim terms associated with genes in *Arabidopsis* and wheat modules. Terms are highlighted if they are shared by corresponding *Arabidopsis* and wheat modules. Modules are ordered by predicted CT phase for each species. Only terms with -Log10$p$ >3 are shown. Wheat W6 and *Arabidopsis* A4 contained no terms above the significance cutoff and so are not shown. Bubble colour indicates the -Log10$p$-value significance from Fisher's exact test and size indicates the frequency of the GO-slim term in the underlying EBI Gene Ontology Annotation database (larger bubbles indicate more general terms).

"generation of precursor metabolites and energy." Co-expressed genes peaking in day-time modules included light-harvesting and light signalling genes as well as *CYP709B3*, which protects the plant from transpiration-triggered salinity stress during the day [38,39].

In dusk-peaking modules A5 and W4, 8 significantly enriched GO-slim terms were shared between *Arabidopsis* and wheat ($p < 0.01$, turquoise colours in Fig 2A, 2D and 2F). Several genes co-expressed in these dusk modules were involved in auxin transport and signalling including the endosomal sorting complex protein *CHMP1A* that ensures proper sorting of auxin carriers (S6 Note) [40]. There was also a significant enrichment for expression of transcripts encoding AP2-EREBP (ethylene responsive) and ARF (auxin responsive) TF superfamilies within the W4 module (S10 Fig). Interestingly, this was followed 2 h later by the expression of genes with AP2-EREBP TFBSs in their promoter region (W5, S11 Fig).

Finally, 2 evening-phased modules W5 and A6 ($r = 0.80$) were enriched for GO-slim terms concerning several metabolic processes (dark blue colours in Fig 2F). Co-expressed orthologs in these 2 modules included *SEVEN IN ABSENTIA2* that regulates ABA-mediated stomatal closure and drought tolerance in *Arabidopsis* [41], and *HYDROPEROXIDE LYASE1* that contributes to responses to insect attack and mechanical wounding [42].

## Components of the core circadian clock in *Arabidopsis* and wheat

We next compared the dynamics of circadian oscillator components in wheat and *Arabidopsis*. Clock gene orthologs belonging to large gene families were detected by phylogenetic analysis (S12–S16 Figs and S7 Table). Overall, wheat circadian clock genes were expressed rhythmically and with a similar phase to their *Arabidopsis* counterparts (Fig 3). However, the free-running rhythms of clock transcripts in wheat had a mean circadian period that was approximately 3.49 h longer than in *Arabidopsis* (27.23 h and 23.74 h, respectively).

*TaLHY* and *TaTOC1* peaked sharply at dawn and dusk, respectively, during the first cycle in constant light, and maintained an >8 h difference in phase throughout the experiment (Fig 3A and 3B). This is consistent with their phasing in *Arabidopsis*. All 3 homoeologs for *TaGI* were robustly rhythmic (BH $q < 0.01$) and peaked at CT7 (Fig 3C). *TaPRR73* transcripts peaked approximately 5 h before *TaPRR37* transcripts, consistent with the phase divergence of *Arabidopsis PRR7* and *PRR3* (Fig 3D and 3E). However, wheat homoeologs *TaPRR59* and *TaPRR95* had similar expression profiles ($R^2 = 0.68$) peaking marginally apart at CT8 and CT10, in between the peak phases of *Arabidopsis PRR9* (CT5) and *PRR5* (CT11) (Fig 3F and 3G). Therefore, the *PRR* gene family in wheat peaks in the order of *TaPRR73*, [*TaPRR37*, *TaPRR95*, *TaPRR59*] in quick succession, and finally *TaTOC1*. This sequential pattern matches the expression of PRR homologs in rice [43].

Transcripts encoding evening complex components, *LUX*, *ELF3*, and *ELF4*, are circadian regulated in *Arabidopsis* and peak simultaneously at dusk. Three wheat triads for LUX-like genes were identified, 1 with higher identity to *LUX/BOA* and 2 similar to other *LUX*-like *Arabidopsis* genes (S15 Fig). Transcripts from all 3 of these triads accumulated rhythmically and peaked from midday to dusk, *TaLUX-Lb* at CT7, *TaLUX/BOA* at CT10, and *TaLUX-La* at CT12 (Figs 3H and 3I and S17). Five wheat transcripts with homology to *Arabidopsis ELF4* and *ELF4-L1-4* accumulated with a mean circadian phase of 12.6 h, similar to *ELF4*, but with lower relative amplitudes (Figs 3J and 3K and S17). *TaELF3-1D* has previously been found to underlie the *Eps-D1* QTL in wheat variety Cadenza [18,19] which is likely to be due to an introgression on chromosome 1D [44]. *Eps-D1* is associated with earlier flowering under long days in wheat, particularly under warm temperatures [45]. Under constant conditions, *TaELF3* transcripts were all arrhythmic (BH $q > 0.36$; Fig 3L). There was no significant difference between the mean expression level across the 3 homoeologs in *TaELF3* when mapped to both Chinese spring and Jagger gene models (F(2, 51) = 2.005, $p = 0.145$, 1-way ANOVA). This suggests that the associated effects of the *TaELF3-1D* allele on heading date in Cadenza

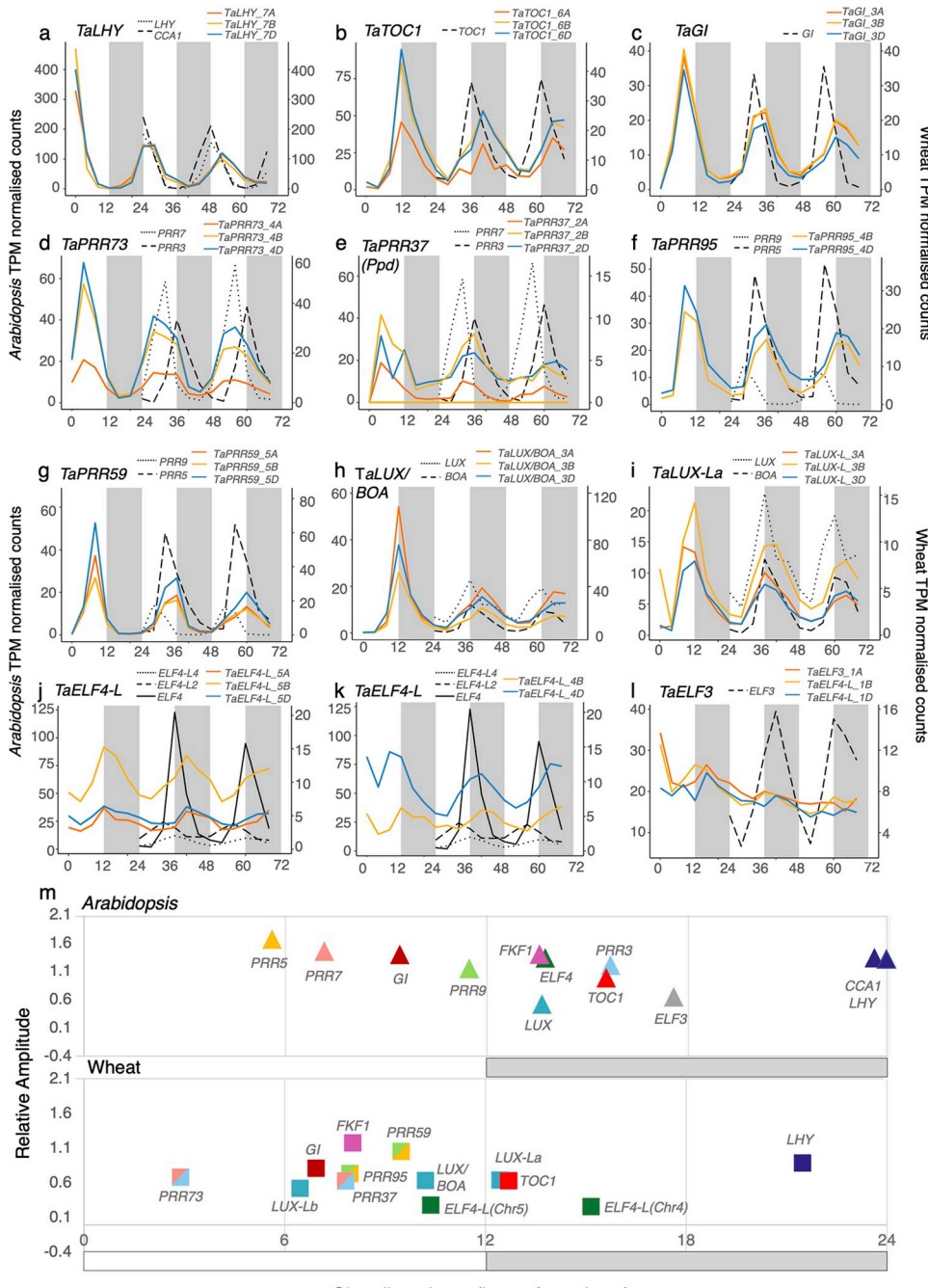

**Fig 3. Free-running expression of core circadian clock genes in wheat and their homologs in *Arabidopsis*. (a–l)** Wheat circadian clock genes were identified through alignment of phylogenetic protein family trees or BLASTP to known clock gene homologs. Gene IDs for each gene set are in S7 Table. Wheat homoeologs are coloured according to their identity to either the A genome (orange), B genome (yellow), or D genome (blue) and grey and white blocks indicate subjective dark and light time periods under constant conditions. Data represent the mean of 3 biological replicates and transcript expression is collapsed to gene level. Expression profiles for additional core circadian clock genes are in S17 Fig. Expression of Ta*ELF3* transcripts were calculated by mapping to both CS and Jagger gene models for the chromosome 1D introgression shared between Jagger and Cadenza. **(m)** Phases of core clock genes in *Arabidopsis* and wheat (meta2d estimates from data 24–68 h after transfer to L:L). Genes were not plotted if BH *q*-values were >0.01. Wheat values represent circular mean circadian phases (CT) across homoeologs calculated in S8 Table. (Data_Fig 3a-m in S1 Data).

are not due to an overall suppression of expression on *TaELF3-1D* relative to the other homoeologs but rather protein-level differences in the introgressed orthologue.

We next assessed the balance of circadian expression between triad homoeologs in the core clock network. *TaLHY*, *TaGI*, and *TaPRR59* had notably similar expression patterns in terms of phase, period, and relative amplitudes over all time points (Fig 3A, 3C, 3F and 3G), suggesting that unbalance in these triads is selected against. *TaLUX/BOA-3A*, *TaLUX-La-3B*, and *TaLUX-Lb-1D* had marginally shorter periods (>2 h) and delayed phases (>2 h) compared to their respective homoeologs (S8 Table).

The REVEILLE family are CCA1/LHY-like MYB-domain TFs that are predominantly activators of evening expressed genes [46,47]. The wheat *RVE* genes could be split into a *LHY* clade (containing the *TaLHY* triad described above), a *RVE6/8*-like clade containing 3 wheat triads and a *RVE1/2/7*-like clade also containing 3 triads (S12 Fig). All *TaRVE6/8* transcripts peaked at CT0-4 concurrently with *TaLHY* (S17 Fig). The *TaRVE2/7* transcripts peaked with distinct phases, *TaRVE27b* in phase with *TaLHY*, *TaRVE27c* 4 h before *TaLHY* and *TaRVE27a* approximately 12 h before *TaLHY* (S17 Fig). Based on their phylogenetic relationships, it is probable that several *RVE2/7* clade paralogs in wheat and *Arabidopsis* arose independently after their evolutionary divergence, and it is therefore interesting that they both show distinct phases of expression, suggesting homoplastic circadian functions.

Expression of orthologs for additional transcripts involved in circadian regulation (*FKF1*, *ZTL*, *LKP2*, *LNK1/2*, *CHE*, and *LWD*) are reviewed in S7 Note and S17 Fig.

## Circadian control of photosystem and light signalling gene expression is largely conserved between *Arabidopsis* and wheat

A further GO-slim analysis across all rhythmically expressed genes in *Arabidopsis* and wheat identified enrichment of similar GO-slim processes including "photosynthesis" ($p < 1 \times 10{-}14$), "rhythmic process" ($p < 1 \times 10{-}6$), "response to abiotic stimulus" ($p < 1 \times 10{-}13$), and "cellular macromolecule biosynthetic process" ($p < 1 \times 10{-}5$, Fisher's exact test, S9 Table). We decided to investigate genes associated with these GO-slim categories and compare expression of their transcripts in wheat versus *Arabidopsis*, acting as case studies to highlight similarities and differences in circadian control between the 2 species. Expression data and Metacycle statistics for all transcripts in this analysis are in S10 Table.

In considering photosynthesis, we examined specifically nuclear genome-encoded photosystem (PS) proteins. Transcripts encoding the PSI components *LHCA1-6*, the PSI reaction centre subunits *PSAD* and *PSAE* and the *PSII* subunits *LHCB1-7* were expressed rhythmically expressed in both species (S10 Table and S18 Fig). In both *Arabidopsis* and wheat *LHCA1-4* peaked towards the end of the subjective day and *LHCA5* and *6* peaked during the subjective night (Fig 4A). Similarly, *LHCB1-6* transcripts peaked in sync with *LHCA1-4* transcripts in both species, and *LHCB7* transcripts had lower relative amplitudes compared to other *LHCB* transcripts (Fig 4B). These analogous relationships suggest that circadian regulation of photosystem protein transcription is conserved in *Arabidopsis* and wheat. One difference was for *PSB27* transcripts that peaked during the subjective day in *Arabidopsis* and during the subjective night in wheat. PSB27 is a protein associated transiently with the PSII complex involved in adaptation to fluctuating light intensities [48] (S18 Fig).

We next investigated expression of transcripts for photoreceptors and light signalling proteins due to their pervasive influence upon development, metabolism, and circadian timing. Although transcripts for the UV-B photoreceptor *UVR8* accumulated with a circadian rhythm in both wheat and *Arabidopsis*, only 1 PHYTOCHROME ortholog (*PHYA)* and 3 *CRYPTOCHROME* orthologs *(CRY1)* were rhythmic in wheat out of 18 orthologs identified (S19 Fig).

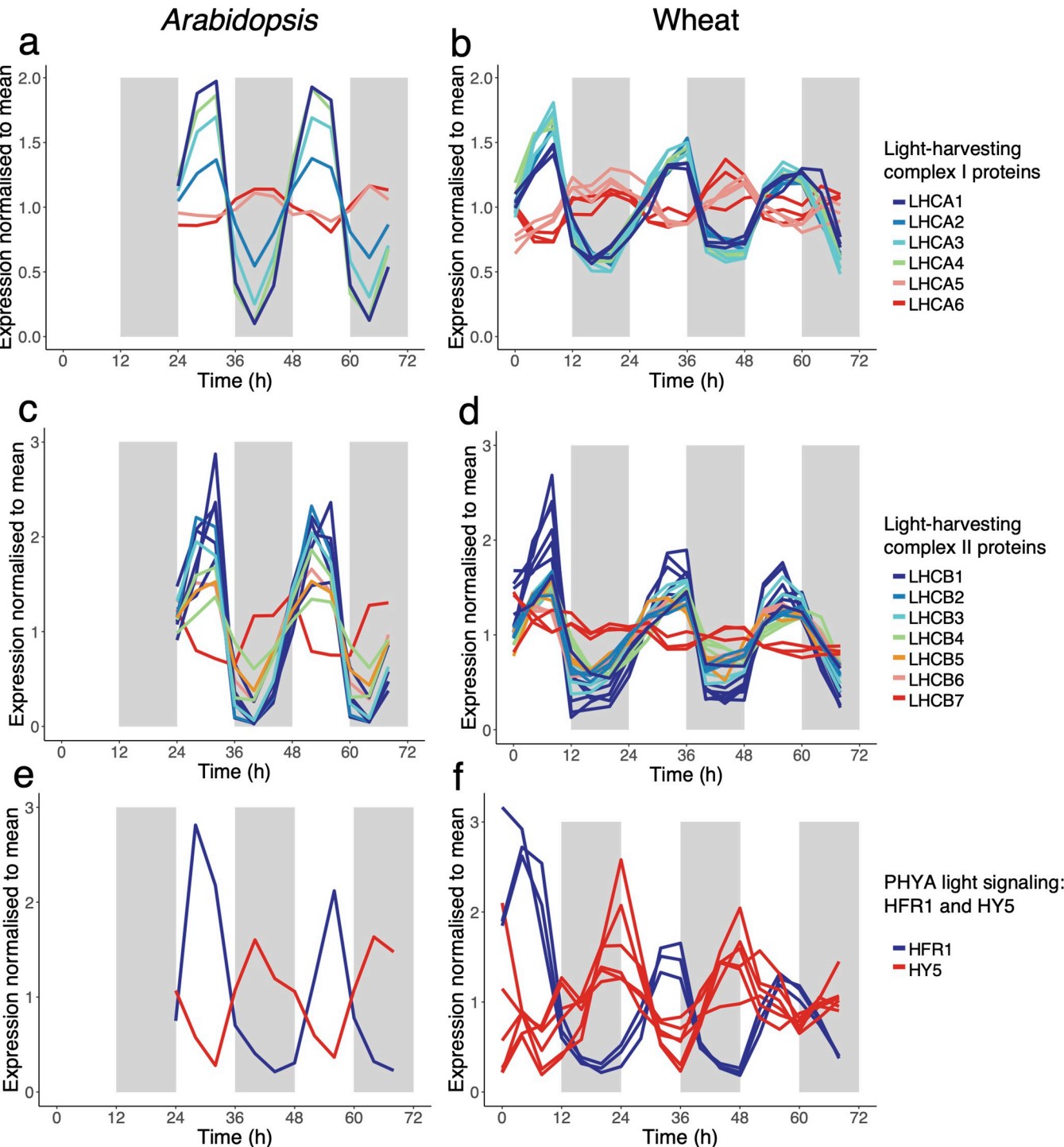

**Fig 4. Conserved expression of circadian regulated transcripts involved in photosynthesis and light signalling.** Examples of transcripts encoding light-harvesting chlorophyll a/b-binding (LHC) proteins associated with photosystem I (*LHCA1-6*) in *Arabidopsis* (a) and wheat (b) and photosystem II in *Arabidopsis* (c) and wheat (d). Two regulators of photomorphogenesis acting downstream of PHYA signalling, *LONG HYPOCOTYL 5 (HY5)* and *LONG HYPOCOTYL IN FAR-RED 1 (HFR1)* are circadian regulated and show antiphase expression patterns in *Arabidopsis* (e) and wheat (f). Transcripts are measured in TPM and are mean normalised to aid the comparison of phase synchrony. Gene IDs of transcripts shown are in S10 Table. (Data_Fig 4a-f in S1 Data).

This contrasts with *Arabidopsis*, where *PHYA-C*, *CRY1*, and *CRY2* accumulated with a circadian rhythm.

Downstream light signalling proteins COP1 and SPA form complexes that degrade positive regulators of photomorphogenesis (e.g., *HFR1* and *HY5*) under dark conditions [49]. Transcripts for *COP1*, *SPA4*, *HFR1*, and *HY5* accumulated rhythmically and with conserved phases in both species (S19 Fig). *COP1/SPA4* peaked synchronously around the end of the subjective night. Surprisingly, given the similar role HFR1 and HY5 proteins have in preventing hypocotyl elongation in low light, *HFR1* and *HY5* transcripts were expressed anti-phase to each other in both species (Fig 4E and 4F). HY5 and HFR1 act synergistically to coordinate the photomorphogenesis response, although it has been suggested that their activation is regulated through independent pathways [50].

Wheat triads with identity to *Arabidopsis PIN1*, *PIN4*, *PIN5*, and *PIF4/5* were rhythmically expressed, alongside 2 triads with high similarity to rice *OsPIL11* and *OsPIL13* [51]; S19 Fig). Overall, we observe that the arrhythmic accumulation of most of the wheat PHY and CRY transcripts is not reflected in the rhythmic expression of several downstream light signalling transcripts. This supports the notion that regulatory signals from photoreceptors might occur at the level of protein stability and localisation rather than at the level of transcript accumulation, as occurs for *ZTL* or *HY5* in *Arabidopsis*.

A set of proteins that link light signalling, circadian regulation and chloroplasts are the sigma factors [52]. These light-responsive nuclear-encoded regulators of chloroplast transcription guide promoter recognition and transcription initiation by plastid-encoded RNA-polymerase (PEP) on the chloroplast genome [53–56]. In *Arabidopsis*, *SIG1*, *3*, *4*, *5*, and *6* were rhythmically transcribed (S19 Fig). In wheat, all homoeologs in triads orthologous to *SIG1*, *SIG3*, and *SIG5* were also rhythmic (BH $q < 0.01$). While the dawn phase of *TaSIG5* transcripts were similar to *AtSIG5*, *TaSIG1* transcripts were expressed over 10 h earlier than *AtSIG1* (S19 Fig). Previous research has shown that activity of *AtSIG1* can be regulated through redox-dependent phosphorylation [57], and activity of all sigma factors are likely to be subject to multiple layers of regulation in addition to circadian control of transcript expression.

## Similarities and differences in circadian control of primary metabolism genes in *Arabidopsis* and wheat

In both *Arabidopsis* and wheat, transitory starch levels increase during the day and are linearly catabolised during the night with the rate of degradation matching the duration of the night [23,58]. Wheat differs from *Arabidopsis* in that carbon storage in the leaves primarily uses sucrose rather than starch; however, starch still accounts for up to 30% of the carbohydrate stored in wheat leaves during the day [58]. Another important metabolite, Trehalose 6 phosphate (Tre6P) is associated with both sucrose regulation and circadian regulation in *Arabidopsis* [59–61] and has been shown to affect grain yield and drought resilience in wheat, maize, and rice [62]. Because of their importance to agriculture and the potential for differential circadian regulation in *Arabidopsis* and wheat, we next compared the circadian expression profiles of genes encoding enzymes that regulate Tre6P and starch metabolism. Overall, we find a surprising number of transcripts that are expressed rhythmically in *Arabidopsis* but are not expressed rhythmically in wheat (coloured in pink, Fig 5) or else are expressed with different peak phases (coloured in yellow, Fig 5). A more detailed analysis of the individual components in this network is described below.

Tre6P synthase (TPS) and Tre6P phosphatase (TPP) participate in the synthesis and dephosphorylation of Tre6P. In *Arabidopsis*, transcripts for *TPS1* (the most well characterised of the T6P synthases), as well as TPS*2*, *6*, *8*, *9*, *10*, and *11* and *TPPA*, *E*, *F*, *G*, and *H* were

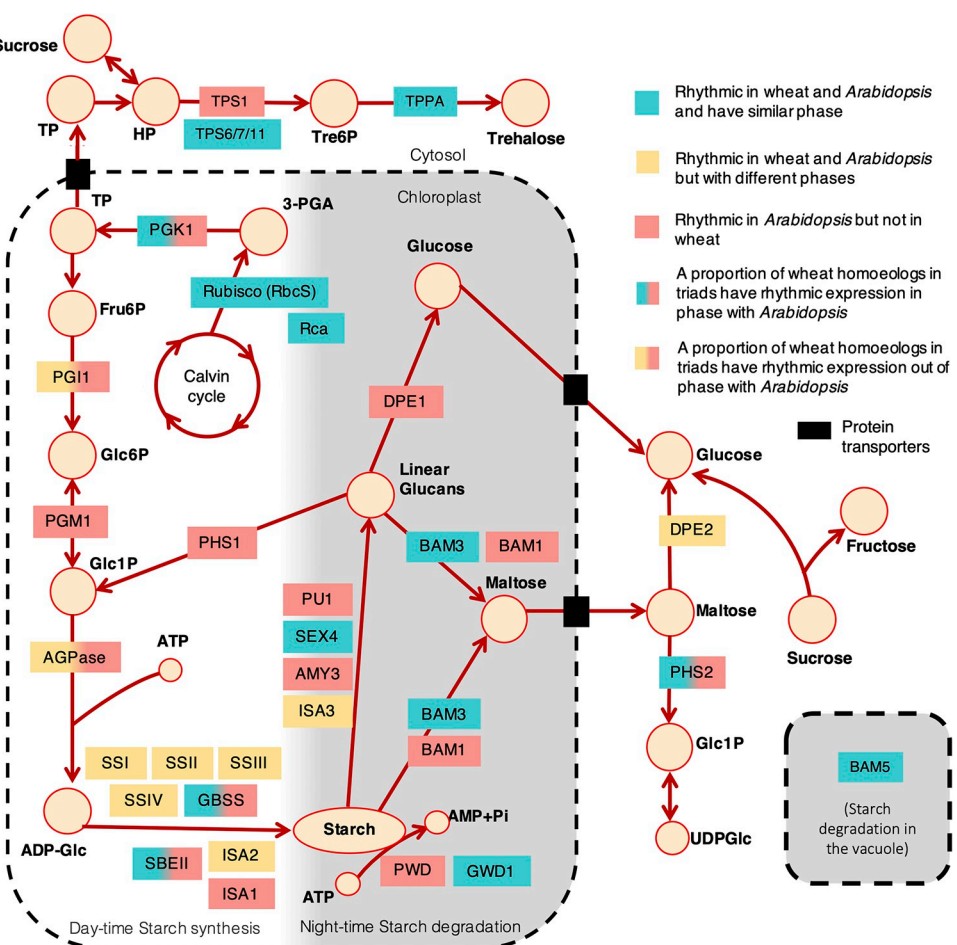

**Fig 5. Similarities and differences in circadian control of transcript accumulation in key genes involved in primary metabolism and signalling.** Circles represent metabolites involved in the breakdown and biosynthesis of starch. Starch synthesis occurs during the day and breakdown occurs at night as indicated by the white to grey shading gradient. The dotted line encloses processes that take place in the chloroplast or vacuole. HP, Hexose-phosphate; Tre6P, Trehalose-6-phosphate; TP, Triose phosphate; 3-PGA, Glycerate 3-phosphate; Fru6P, Fructose-6-phosphate; Glc6P, Glucose-6-phosphate; Glc1P, Glucose-1-phosphate; ATP, Adensoine tri phosphate; ADP-Glc, ADP-glucose; TPS, Trehalose phosphate synthase; TPP, Trehalose phosphate phosphatase; PGK1, Phosphoglycerate kinase 1; PGI1, Glucose-6-phosphate isomerase; PGM1, Phosphoglucomutase-1; PHS1 and 2, ALPHA-GLUCAN PHOSPHORYLASE 1 and 2; AGPase, ADP-Glc pyrophosphorylase; BAM1,3,5, β-amylase 1,3,5; ISA1,2,3, Isoamylase 1,2,3; DPE1,2, Disproportionating enzyme1 and 2; SBEI,II, Starch branching enzyme I, II; PU1, Pullulanase 1; PWD, Phosphoglucan, water dikinase; GWD, α-glucan, water dikinase; SEX4, starch excess 4; AMY3, α-amylase; GBSS, Granule bound Starch synthase; SSI-IV, Starch synthase I-IV; UDPGlc, UDP-glucose. Pathway references: [116–118].

expressed rhythmically (S20 Fig). Wheat transcripts for *TPS1* were arrhythmic; however, rhythmic transcripts were found in triads more closely related to TPS11, 6 and 7 (Figs 5 and S20). We identified 3 rhythmic TPP triads in wheat, 2 of which were orthologous to *Arabidopsis TPPA, F* and *G*. The third TPP triad was part of a monocot-specific clade [62] that includes *Zm00001d032298*, a crop improvement target in maize.

Ribulose bisphosphate carboxylase (Rubisco) comprises 8 small (RbcS) and 8 large (RbcL) subunits, which are encoded by the nuclear and chloroplast genomes, respectively [63]. Rubisco requires activation by Rubisco activase (RCA) to release its activity from inhibitory substrates [64]. In our wheat expression data, 22 putative wheat orthologs for the small subunit of Rubisco were rhythmic, peaking during the subjective night, as *RBCS1A, RBCS1B, RBCS2B,*

and *RBCS3B* do in *Arabidopsis* (Figs 5 and S20). Two triads with identity to RCA were identified, one of which accumulated rhythmically (peaking at CT0, as with *Arabidopsis* RCA).

Circadian regulation has a pervasive influence on starch metabolism in *Arabidopsis*, particularly the nocturnal rate of transitory starch degradation [23,65]. Chloroplast phosphoglucose isomerase 1 (*PGI1*) and chloroplast phosphoglucose mutase (*PGM1*) are essential enzymes that link the Calvin–Benson cycle with starch biosynthetic pathway [66–68]. In *Arabidopsis*, these transcripts accumulated with a circadian rhythm (BH $q < 1 \times 10^{-4}$); *PGM1* peaked just after dusk (CT14), and *PGI1* slightly later at CT20. In contrast, only 1 wheat *TaPGI1* homoeolog was rhythmic (BH $q < 0.01$), which had a low relative amplitude (0.16) and a peak phase of CT8. No homoeologs for *TaPGM1* were rhythmically expressed (BH $q > 0.01$, S20 Fig).

ADP-glucose pyrophosphorylase (AGPase) mediates the first irreversible and rate-limiting step in starch biosynthesis through the formation of ADP-Glc. In *Arabidopsis*, transcripts encoding the small and large subunits of AGPase (*APL1*, *APL2*, *APL3*, *APS1*) were rhythmic, peaking at night around CT20. In comparison, in wheat only 2 of the 11 transcripts with homology to *APL1*, *APL2*, and *APS1* were rhythmic (BH $q < 0.01$), with the remaining transcripts lacking a discernible rhythm (BH $q > 0.05$) (Figs 5 and S20).

Starch synthases (SS) represent another group of metabolically important enzymes that use the glucose from ADP-Glc to elongate glucan chains. In *Arabidopsis*, there are 5 types: SSI, SSII, SSIII, SSIV, and granule bound GBSSI. SSI-IV are responsible for synthesis of amylopectin, with SSIII and IV determining starch granule number and morphology [69]. *GBSSI* is a known dawn-expressed gene, regulated directly by CCA1/LHY, specialised for amylose synthesis [70]. In wheat, *GBSSI* orthologs are called *TaWaxy* and cultivars with 3 null alleles produce amylose-free starch in their grain [71]. Comparison of starch synthase expression in *Arabidopsis* and wheat revealed several differences between the phases and relative amplitudes of these transcripts (Figs 5 and S20). In *Arabidopsis*, *GBSSI* transcripts had by far the greatest relative amplitude (1.26) with peak expression at dawn. The next greatest amplitudes were of *SSIV* transcripts, which peaked at CT17. *SSII* and *SSIII* peaked together at CT21 and *SS1* peaked at CT8 with a much smaller amplitude (0.12). In contrast, in wheat, an *SSIII* triad (*TaSSIIIb*) had the largest relative amplitude rhythms of the wheat starch synthases identified (0.64 to 0.73). Wheat transcripts for *SSI* and *SSIV* also peaked in the morning, whereas wheat *SSII* peaked instead in the subjective night (~CT15). In our data, *TaWaxy* (*GBSSI*) transcripts were present at a very low baseline level (<0.01 TPM) and without any circadian oscillation. However, another wheat triad, *TaGBSSII*, shared >62% identity with *Arabidopsis GBSSI*, and the B and D homoeologs had rhythmic expression which peaked at dawn. *TaWaxy* and *TaGBSSII* are specific to endosperm and leaf tissues, respectively [72], which might explain the distribution of transcript accumulation seen here. We can conclude that the circadian clock regulates the expression of SS transcripts in both *Arabidopsis* and wheat, although there might be an emphasis on different types of SS in each species.

The *Arabidopsis* circadian clock regulates the rate of starch degradation so that starch reserves are depleted precisely at subjective dawn [23]. Many transcripts encoding starch-degrading enzymes in *Arabidopsis* had synchronised dusk peaks: isoamylase-type starch debranching enzyme *ISA3*; alpha-amylase *AMY3*; plastidial phosphorylase *PHS1-2*; disproportionating enzymes *DPE1-2*; glucan, water dikinases *GWD1* and *PWD* and glucan phosphatase *SEX4*. *Arabidopsis* transcripts for *BAM3*, *BAM5*, and *PU1* also oscillated with a circadian rhythm, peaking later during the subjective night. Strikingly, wheat orthologs for several of these genes were not rhythmic, including *AMY3*, *DPE1*, *PWD*, *PHS1*, *PU1*, and *BAM1*. Wheat orthologs for *ISA3* and *DPE2* were expressed rhythmically, but peaked approximately 8 to 12 h ahead of their *Arabidopsis* counterparts. Some starch degradation enzymes had conserved circadian expression patterns in the 2 species, such as *SEX4*, *GWD1*, *BAM3*, and *BAM5*

transcripts. GWD catalyses glucan phosphorylation and *SEX4* encodes a phosphoglucan phosphatase, both of which facilitate hydrolytic attack by β-amylases (BAM) in the early steps of starch degradation [65,73].

## Discussion

### Conservation of circadian regulation between homoeologous genes

We identified a large proportion of imbalanced circadian triads in our dataset. It was our initial expectation that there would be strict balance between the majority of circadian regulated homoeologs due to the critical and finely balanced role the clock has in regulating the transcriptome and due to the reported high levels of balance reported from single time point transcriptomic analysis in wheat [32]. Instead, we find 3 times as many triads with imbalanced circadian rhythms as triads with balanced circadian rhythms. This is likely to be due in part to our classification of circadian imbalance as any triad with different rhythmicity, period, phase, or relative amplitudes between homeologs. Another factor that distinguishes ratios of transcriptional balance (as defined by Ramírez-González and colleagues [32]) and circadian balance is that transcriptional balance within circadian triads is often dynamic across a time course, shifting between balanced, dominant and suppressed relationships over time (S6 Fig).

The imbalance of many of these circadian triads was due to arrhythmicity in 1 or 2 homoeologs that were expressed at a lower mean level than the rhythmic homeologs. This reduction of expression could be due to constitutive epigenetic silencing or changes to promoter regions, allowing differential binding of TFs [74–76]. These are likely to be triads where 1 or 2 homoeologs take responsibility for performing the biological function of the triad, while the other homoeolog has reduced functionality. We found additional circadian unbalance in the form of altered phase, period, and relative amplitudes. It is possible that some of these differences are due to retention of circadian regulation from the ancestral genome of each homeolog (Fig 1A), although it is likely that other differences reflect more recent diversification in expression as a step towards neo-functionalization. It has been previously suggested that functional divergence is a likely fate for duplicated genes in a sufficiently large population [77]. In *B. rapa*, 42% of circadian controlled paralogs had differential expression patterns [31]; however, these paralogs arose through whole-genome duplication events around 13 to 43 million years ago, so have been exposed to longer periods of time during which selection could act upon these duplicate genes [78]. In comparison, specialisation of circadian homeologs in wheat could be comparatively lower due to the relative infancy of its polyploidisation around 10,000 years ago.

### Differences between periods of rhythmic transcripts in *Arabidopsis* and wheat

The mean period of circadian regulated genes in wheat was over 3 h longer than in *Arabidopsis*. Period length is affected by a range of exogenous conditions (e.g., light and temperature) and varies between tissues and plant age [79]. There is also evidence that longer periods have been selected for during cultivation of crops at higher latitudes [1,2,80], potentially due to enhanced seasonal tracking capability enabling precision timing of growth and flowering [81]. Compared to other plant circadian transcriptome data, rhythmic wheat transcripts also had higher period variance (Fig 1C). We have several hypotheses for why this might be the case. The broad period distribution in wheat might be a technical effect from inclusion of all aerial plant material in our sampling strategy. Variation in free-running periods could occur at the organ-, tissue-, or cellular-level, and transcripts that are highly expressed in those regions may reflect those period differences [82,83]. An alternative possibility is that period variation is due

to uncoupling of multiple circadian oscillators within the same cell that control expression of subsets of transcripts [84–86]. A third possibility is that the high variability of periods in wheat may be a product of polyploidisation, as functional redundancy between homoeologs may allow a tolerance for less tightly regulated, non-dominantly expressed circadian transcripts. Future research could examine the relationship between period distributions of circadian transcriptomes and the effects of domestication, latitudinal adaptation, monocot-dicot divergence, and polyploidy. It would also be interesting to investigate cellular heterogeneity in circadian expression across different tissues to see how well rhythms are synchronised.

## Similarities and differences in circadian regulation between wheat and *Arabidopsis*

Our analysis revealed extensive conservation of time-of-day specific GO-slim processes and co-expressed genes between *Arabidopsis* and wheat. These included genes involved in photosynthesis (e.g., photosystem proteins), light signalling (e.g., *HFR1*, *HY5*, *PINs*), translation (e.g., ribosome proteins) and auxin and ethylene responsive TFs. The striking conservation of photosynthesis-related genes was also reflected by the enrichment of these genes in both balanced wheat triads and similarly expressed circadian paralogs in *Brassica rapa*. Photosynthetic outputs have also been reported to be governed by the circadian clock in the liverwort species *Marchantia polymorpha*, in diazotrophic cyanobacterium *Cyanothece* sp. and in alga *Aegagropila linnaei*, perhaps suggesting a widespread control mechanism with an ancient evolutionary origin [87–89].

We also identified several interesting differences between *Arabidopsis* and wheat, including absence of rhythmicity in wheat *PHY* and *CRY* transcripts and antiphase expression of the wheat sigma factor *SIG1*. Furthermore, we found differences in rhythmic expression of many transcripts involved in regulating Tre6P and starch metabolism.

In our data, putative wheat homeologs of *TPS1* were arrhythmic. Instead, rhythmic *TPS* transcripts in wheat had similarity to *Arabidopsis TPS11*, *6* and *7* (S20 Fig). In *Arabidopsis*, TPS1 is the most catalytically active and best characterised TPS and feeds back into the entrainment of the circadian clock [60,90]. If the lack of rhythmicity in wheat *TPS1* transcripts is reflected at the level of protein activity, it may indicate that Tre6P synthesis is not regulated as tightly by the circadian clock in wheat as in *Arabidopsis*. On the other hand, circadian control of other TPS triads may have implications for biotic or abiotic defence in wheat. TPS5-11 has been previously implicated in control of stomatal aperture [91], thermotolerance [92], and defence against fungal, bacterial, and aphid attack [93,94]. In rice, *OsTPS8* influences drought resistance through suberin deposition [95], and wheat *TaTPS11* participates in a cold stress response [96].

In wheat, transcripts for starch degradation enzymes (*PHS1*, *DPE1*, *BAM1*, *PU1*, *AMY3*, *PWD*) and starch biosynthesis enzymes (*PGI1*, *PGM1*, *ISA1*, and *ATPase*) had either arrhythmic expression or low relative amplitudes compared with the robust rhythms of many of these transcripts in *Arabidopsis*. Additionally, *ISA2*, *ISA3*, and the starch synthases (*SSI-IV*) had differing circadian phases between the 2 species. While it is possible that rhythmic expression of a reduced number of genes (e.g., *SEX4*, *GWD1*, *BAM3*, and *BAM5*) is sufficient to mediate circadian control of starch degradation in wheat, these data suggest that the circadian clock has a less pervasive influence upon transcriptional control of starch metabolism in wheat compared to *Arabidopsis* (S8 Note). This might reflect the reduced role of transitory starch as a storage carbohydrate in wheat leaves. Future studies could assess in more detail circadian regulation of sucrose and fructan metabolism, as these are known to have a greater role in carbon assimilation and adaption to freezing tolerance in wheat [58,97].

## Conclusions

Our data reveal the influence of circadian regulation on the wheat transcriptome and highlight several intriguing differences between rhythmically expressed transcripts in *Arabidopsis* and wheat. It explores the added complexity of coordinating circadian expression across multiple subgenomes in a hexaploid species. Given the circadian clock has been under selection during domestication and presents multiple targets for crop improvement, it is likely that this new insight into the clock in wheat will be important in the development of new sustainable and resilient cultivars. It is our hope that these data provide a resource for identifying target genes regulated by the circadian clock, allowing the relationships between chronotype, yield and resilience to be explored in future studies.

## Methods

### Plant materials and growth conditions

Wheat: Wheat seeds cv. Cadenza were imbibed for 3 days on damp filter paper on a Petri dish at 4°C. Cadenza is a UK hexaploid spring wheat that is the background variety for a mutagenized population of TILLING lines [98]. Plates were moved at dawn (06.00 = CT0), to a growth cabinet set to 22°C under 12:12 light: dark cycles (approximately 200 μmol m−2 s−1). After 2 days, only seedlings with fully emerged radicles were sown, 3-cm deep in Petersfield cereal mix in 9-cm pots. Plants were not vernalized. Seedlings were grown under 12hlight:12hdark conditions for 14 days. After 14 days, at dawn, seedlings were transferred to constant light conditions and tissue was sampled every 4 h for 3 days (CT0-CT68; 54 samples in total, across 18 time points). At each time point, we sampled the entire aerial tissue from 3 replicate plants, which was frozen immediately in liquid nitrogen before storage at −80°C. Total RNA was extracted using Qiagen Rneasy plant mini kits (cat. No. 74904) with on-column DNAse treatment (RNAse-Free Dnase Set (cat. No. 79254). RNA concentration and integrity were quantified using a Nanodrop Spectrophotometer and Perkin Elmer LabChip GX Nucleic acid analyser before sequencing.

Details of growth conditions for *Arabidopsis* [26], *Brassica rapa* [31], *Brachypodium distachyon* [29], and *Glycine max* [30] datasets can be viewed in their source manuscripts. Briefly, all circadian data were measured under constant light and temperature following 12h:12h light:dark entrainment other than *Glycine max* [30] that was entrained under 16h:8h light:dark cycles.

### Wheat mRNA sequencing, read alignment, and quantification

Library preparation was carried out following the Illumina TruSeq protocol and reads were sequenced on a NovaSeq S2 flow cell at the Earlham Institute. Approximately 150-bp paired-end reads were generated from each library to an average depth of 84 M reads per replicate. Reads were filtered for quality and any remaining adaptor sequence was trimmed with Trimmomatic [99]. Surviving reads were aligned to the Chinese Spring RefSeq v1.1 wheat genome [11] using HISAT2 [100] with default parameters. Uniquely mapping reads were then quantified using StringTie [101] and TPM values were extracted for each gene per sample. Data presented are quantified at gene-level TPM (summed across individual transcripts) and are averaged across 3 biological replicates.

A previously described telomeric introgression on chromosome 1D contains an introgressed Ta*ELF3* orthologue [44]. To account for mapping bias and accurately characterise the expression of this gene (Fig 3L), filtered and trimmed mRNA reads were aligned as above but to a pseudo reference constructed by concatenating Chinese Spring RefSeq v1.0 to the well-

defined 1D introgression extracted from the Jagger genome assembly [102]. TPM values were summed across each pair of introgressed 1D and Chinese Spring 1D orthologues.

## Processing and quantification of previously published datasets

Raw reads from previously published circadian datasets were downloaded for *Arabidopsis* [26], *Brassica rapa* [31], and *Brachypodium distachyon* [29]. These reads were filtered for quality, and any remaining adaptor sequence trimmed with Trimmomatic [99]. Surviving reads were aligned using HISAT2 [100] to *A. thaliana* genome (TAIR 10), *B. rapa* genome (v1.0), and the *B. distachyon genome* (v3.0), respectively. For the *Arabidopsis* alignment, maximum intron length was set to 5,000 nt consistent with pre-processing in [26,103]. StringTie [101] was used to quantify uniquely mapping reads before TPM value extraction at gene level. For *Glycine max* [30], FPKM normalised reads were downloaded from the *Glycine max* RNA-seq Database [104] (accession GSE94228) and were converted from FPKM to TPM prior to analysis.

## Homolog identification of circadian clock and circadian controlled genes

Wheat homologs of *Arabidopsis* core circadian clock genes were identified in the wheat genome by detecting similarity to the following conserved protein family domains that are present in the proteins encoded by these genes: MYB1R, a subtype of MYB domain that contains a distinctive SHAQKY sequence motif (present in the CCA1, LHY, and RVE[1–8]) or a distinctive SHLQKY sequence motif (present in LUX), PAS (present in ZTL), PRR (present in TOC1 and PRR[3579]), and ELF4 (present in ELF4). A hidden Markov model (HMM) for each domain was used in HMMER 3.2.1 HMMSEARCH [105] to search for members of the domain family in the following proteome datasets: Araport11 (*Arabidopsis thaliana*), RGAP7 (*Oryza sativa*), JGI Phytozome version 12 (*Brachypodium distachyon*), IBSC (*Hordeum vulgare*), SpudDB PGSC v4.03 (*Solanum tuberosum*), and IWGSC Refseq v1.1 (*Triticum aestivum*). The HMMs provided by Pfam (https://pfam.xfam.org/) were used for the PAS domain (PAS_9, PF13426), the PRR domain (Response_reg, PF00072), and the ELF4 domain (PF07011). For the MYB domain, an HMM was built for the MYB1R subfamily using HMMER3 HMMBUILD [105] with an alignment of protein sequences from *Arabidopsis* and rice, previously established as being members of this subfamily. The sequences found from these genomes were realigned to the original alignment using HMMER 3.2.1 HMMALIGN [105]. Amino acids with non-match states in the HMM were removed from the alignment and alignment columns with <70% occupancy were also removed. The longest splice variant of each protein was selected to estimate a phylogenetic tree with bootstrap support using RAxML 8.2.12 [106] with the following method parameters set: -f a, -x 12345, -p 12345, -# 100, -m PROTCATJTT. The trees were mid-point rooted and images created using the Interactive Tree of Life (iToL) tool [107]. For the larger MYB and PRR families, proteins from the tree clades containing known clock gene(s) were realigned across their full-length and a "nested" phylogenetic tree was re-estimated with RAxML as described above. The tree was visualised in the iTOL website alongside the corresponding alignment. This view provided increased detail about the relationships within the clade and enabled orthologous sequences to be inferred. Wheat homologs for *ELF3*, *GI*, *LWD1/2*, *CHE*, and *LNK1/2* were identified by BLASTP searches using previously identified wheat and *Brachypodium* predicted proteins confirmed by reciprocal BLAST searches against *Arabidopsis*. IDs and source references can be viewed in S7 Table.

Putative wheat orthologs for *Arabidopsis* circadian controlled pathway genes involved in photosynthesis, light signalling, and primary metabolism were first extracted using Biomart

v0.7 [108] available from Ensembl Plants and taken forward if they had >40% identity in the DNA sequence. Orthologs were then verified using BLASTP using *Arabidopsis* protein sequences as a query against the wheat protein database to confirm the wheat gene IDs. Complete lists of wheat gene IDs used in the pathway analysis can be viewed in S10 Table.

### Circadian quantification using Metacycle and Biodare2

To estimate proportions of rhythmic genes expressed in *Arabidopsis* and wheat, we removed only genes with 0 TPM at all time points. This approach has been used in several previous studies [26,109,110] and allows detection of low-expression rhythmic transcripts. An analysis of how filtering for low-expression genes affects the estimates of proportions of rhythmically expressed genes is discussed in S1 Note and S1 Table.

The R package MetaCycle [27] was used to identify rhythmically expressed transcripts (BH $q$-values) and to quantify period lengths (hours), absolute phase (hours), baseline expression (TPM), amplitudes (TPM), and relative amplitudes of circadian waveforms. Relative amplitude is the ratio between amplitude and baseline TPM if the baseline is greater than 1. Metacycle integrates results from 3 independent algorithms (ARSER, JTK_CYCLE, and Lomb-Scargle) to produce summary "meta2d" statistics that combine the outcome from these algorithms. Metacycle was run using the following parameters; minper = 12, maxper = 35, adjustPhase = "predictedPer." Transcripts were defined as rhythmic if they had $q$-values < 0.05 and high confidence rhythmic transcripts if they have $q$-values < 0.01. To calculate circadian phase (CT; relative to period length = 24), meta2d phase estimates were multiplied by 24 and then divided by the period estimates for each transcript. Circular phase means were calculated using the package "circular" implemented in R [111].

There are many different algorithms available for quantification of rhythmicity within time-series data, some of which perform better on datasets with higher levels of noise, non-24 h periods, or various sampling frequencies. To validate the meta2d results, we also used the FFT-NLLS and MESA algorithms implemented in Biodare2 to verify our observations about period, phase, and rhythmicity [28]. FFT-NLLS also provides relative amplitude error (RAE) statistics that represent a useful metric for assessing rhythmic robustness. FFT-NLLS and MESA were run using the BH $q$ < 0.01 filtered transcripts categorised in Metacycle and with the following parameters: no dtr, min-max, p(12.0–35.0).

To enable as close a comparison with the *Arabidopsis* dataset as possible, the wheat time course was cropped to a data window of 24 to 68 h for approximation of period, phase, and relative amplitude unless specified otherwise. This data window also ensures that measurements are being made under circadian conditions following transfer to constant light. For the triad analysis, meta2d estimates were measured over the full time course (0 to 68 h) as differentiation of homeolog behaviour was the main interest, including the response to transfer to L:L.

### Clustering of rhythmic genes into expression modules

Gene co-expression analysis was carried out using the R package WGCNA (Langfelder and Horvath, 2008; R version 3.6.0.). Parameters used for the network construction are available from our group's GitHub repository: https://github.com/AHallLab/circadian_transcriptome_regulation_paper_2022.

*Arabidopsis*: The 10,317 genes identified by MetaCycle as significantly rhythmic ($q$-value < 0.01) were filtered and genes with greater than 0.5 TPM average expression at more than 3 time points were retained for further analysis. The average expression at each time point for the remaining 10,129 genes was used to construct signed hybrid networks on a replicate basis using the blockwiseModules() function. The soft power threshold was calculated as

18, and the following parameters were used: minModuleSize = 30, corType = bicor, maxPOutliers = 0.05. As a final step in the construction of our networks, we specified a mergeCutHeight of 0.15 for this function. This metric cuts the dendrogram of clustered module eigengenes, using the dissimilarity given by one minus their correlation (here a mergeCutHeight corresponds to a correlation of 0.85) and merges all the modules on each branch. Highly connected hub genes were identified for each of the 9 co-expression modules using the function chooseTopHubInEachModule() and eigengenes were identified for each module using the moduleEigengenes() function.

Wheat: The 18,633 genes identified by MetaCycle as significantly rhythmic across 12 time points CT24—CT68 ($q$-value < 0.01) were filtered and genes with greater than 0.5 TPM average expression at more than 3 time points were retained for further analysis. The average expression at each time point for the remaining 16,327 genes was used to construct signed hybrid networks using the blockwiseModules() function. A soft power threshold of 18 was used, together with the following parameters: minModuleSize = 30, corType = bicor, maxPOutliers = 0.05, mergeCutHeight = 0.15. Eigengenes were identified for each module using the moduleEigengenes() function. For this dataset, we merged modules with closely correlated eigengenes were using the mergeCloseModules() function, using the parameters: cutHeight = 0.25 (corresponding to a correlation of 0.75, iterate = F) and new module eigengenes were calculated for the resulting 9 modules.

## Cross-correlation analysis

A cross-correlation analysis was used to find the shift in time (lag) that produced the highest (peak) correlation between 2 rhythms. This approach was used to identify modules that peaked synchronously (had a peak lag of 0 h) or asynchronously (had a peak lag of 4, 8, or 12 h) by correlating eigengenes for each module (S9 Fig). We also used cross-correlation to identify imbalanced phases within rhythmic triads (Fig 1E). Before calculating the cross-correlation between 2 expression rhythms, we first scaled both expression patterns using their means and standard deviations, so the output reflects a time-dependent Pearson correlation coefficient ranging between −1 and 1:

$$Z_A = \frac{X_A - \bar{X}_A}{S_A}, Z_B = \frac{X_B - \bar{X}_B}{S_R}.$$

Where $Z_i$, $X_i$, $\bar{X}_i$, and $S_i$ represent the standardised expression level, tpm expression level, mean expression level, and standard deviation of gene A and B, respectively. Once both expression patterns have been scaled, the discrete cross-correlation between the 2 expression patterns is calculated using the np.correlate function and is divided by the number of time points in the expression signal returning the Pearson correlation coefficient at different lags. The index of the array with the largest Pearson correlation coefficient score corresponds to the lag that maximises the phase similarity between the 2 temporal expression patterns. Code for the cross-correlation analysis is available from our group's GitHub repository: https://github.com/AHallLab/circadian_transcriptome_regulation_paper_2022.

## Mean-normalised data for oscillation plots

Oscillation plots in S18–S20 Figs were mean normalised to aid visualisation of period and phase differences between transcripts. Data was adjusted by dividing the TPM values at each time point by the mean across all time points for each gene so that the baseline expression was equal to 1.

## Gene ontology term enrichment

Functional enrichment of differentially expressed genes for biological processes within each module was performed using the gene ontology enrichment analysis package, topGO [112] in R (version 3.6.0, with the following parameters: nodeSize = 10, algorithm = "parentchild," classicFisher test $p < 0.05$). Enrichment of terms in all rhythmic genes in *Arabidopsis* and wheat was compared against a background "gene universe" of all expressed genes in each dataset (26,392 genes for *Arabidopsis* and 86,567 for wheat). This gene universe was also used in the GO-slim analysis for enrichment in circadian balanced versus imbalanced triads. Enrichment of terms in expression modules was compared against a background of all rhythmically expressed genes (BH $q < 0.01$) that clustered into modules in each dataset (10,129 genes for *Arabidopsis* and 16,327 for wheat). GO-slim terms refer to ontology terms for biological processes unless otherwise specified and were obtained from Ensembl Plants 51 [11] using the BioMart tool. The bubble plot was plotted using ggplot in R adapting code from [113].

Enrichment of GO-slim terms in *B. rapa* circadian paralogs with similar and differential expression patterns was conducted using previously published DiPALM results for pattern change in a LDHC circadian time course [31]. Using the DiPALM measure of pattern change between paralogs (where a score of 1 is a very similar pattern), we selected paralog pairs with a $p$-value of <0.001 as having differential patterns and pairs with a pattern change $p$-value of >0.1 were considered to be similar patterns. Differential *Brassica* paralogs had a mean Pearson correlation coefficient of 0.31 (SD = 0.43), similar *Brassica* paralogs had a mean Pearson correlation of 0.75 (SD = 0.23). Applying this cutoff also gave us similar numbers of paralog pairs in each category, making it easier to conduct a GO-analysis. In this analysis, we ignored differences in expression change for consistency with our wheat triad analysis. Data was first filtered for rhythmicity using Metacycle $q$-values <0.01. Only paralog pairs with 2 significantly rhythmic paralogs were retained for the GO-slim analysis. Enrichment of terms in similarly expressed circadian paralogs (1,562 genes) or differentially expressed paralogs (1,438 genes) in *B. rapa* was compared against a background of 4,646 genes expressed in the Greenham and colleagues [31] dataset in paralogs and which had GO-slim annotation available.

## Enrichment analysis of transcription factor superfamilies in wheat co-expression modules

Genes annotated as members of TF superfamilies [32] were identified in each co-expression module and the frequency of each TF superfamily compared to the frequency observed in the 16,327 genes submitted to WGCNA. TF families were classed as either significantly under or overrepresented in each module using Fisher's exact test ($p < = 0.05$).

## Enrichment analysis of transcription factor binding sites in wheat co-expression modules

Approximately 1.5 kb of sequence upstream of the transcription start site (TSS) was extracted for each of the 16,327 genes submitted to WGCNA. FIMO, from the MEME tool suite (v 4.11.1) was used to predict TFBS in these regions based on similarity with previously DAP-seq validated TFBS identified in *Arabidopsis* [114]. FIMO was run as reported in Ramírez-González and colleagues [32] ($p$-value threshold of <1e-04 (default),—motifpseudo set to 1e-08 as recommended for use with PWMs and a—max-stored-scores of 1,000,000). The background model was generated from the 16,327 promoter sequences using MEME fasta-get-markov. As the significance of multiple matches of a single TFBS family in the putative promoter region for each gene is unknown, we derived a nonredundant (nr) list of matched TFBS motifs for

each gene within each of the 9 modules and for the complete set of 16,327 genes, where multiple occurrences of a TFBS superfamily in a single promoter sequence were only counted once. The frequency of these nrTFBS motifs for each co-expression module was compared to the frequency of nrTFBS seen across all 16,327 genes and families significantly under or overrepresented in each module were identified using Fisher's exact test ($p < = 0.05$).

## Loom plots

Genome position data for the plots are based on annotations from Chinese spring (Triticum_aestivum.IWGSC.52.gtf) downloaded from Ensembl Plants. Code for creating Loom plots (S8 Fig) is implemented in R and a code package with accompanying R markdown notebooks is available from our group's GitHub repository https://github.com/AHallLab/circadian_transcriptome_regulation_paper_2022.

## Statistical analysis

Statistical tests including Welch's 2 sample *t* test, 2-proportions z-test, 1-way ANOVA, 2-level, nested ANOVA, and Chi-square tests of independence were all conducted in the R "stats" package (version 4.0.0) with default parameters.

## Supporting information

**S1 Note. Proportions of rhythmic genes in expression datasets.**
(DOCX)

**S2 Note. Mean period lengths in rhythmic wheat transcripts over sliding windows of circadian experiment.**
(DOCX)

**S3 Note. Distribution of phases across different period bins for rhythmic wheat transcripts.**
(DOCX)

**S4 Note. Examples of TF triads with imbalanced regulation.**
(DOCX)

**S5 Note. Detecting patterns of triad circadian balance.**
(DOCX)

**S6 Note. Orthologs co-expressed in similar phased modules of gene expression.**
(DOCX)

**S7 Note. Orthologs of further circadian clock components.**
(DOCX)

**S8 Note. Considerations for the differences in circadian regulation observed between *Arabidopsis* and wheat.**
(DOCX)

**S1 Fig. Correlation of wheat period, circadian phase, and absolute amplitude values estimated from meta2d or FFT-NLLS.** Meta2d was run in Metacycle and FFT-NLLS in Biodare2 using 24–68 h data filtered for rhythmicity B.H q-values <0.01. (Data_Fig_S1a-c in S2 Data).
(PDF)

**S2 Fig. Period estimates for rhythmically categorised data predicted using either Metacycle or Biodare2.** MESA and FFT-NLLS are independently run in Biodare2 (A and B), ARSER, JTK, and LS (C–E) are all run through Metacycle to produce an average period prediction meta2d (F). Data for *Arabidopsis* (blue) and wheat (yellow) was filtered for BH q < 0.01 on a data window of 24–68 h after transfer to constant light. (Data_Fig_S2a-f in S2 Data). (PDF)

**S3 Fig. Phase estimates for highly rhythmically categorised data from either Metacycle or Biodare2.** Phases were adjusted by period length so that 1 complete cycle is equal to 24 h. MESA and FFT-NLLS are independently run in Biodare2 (A and B, G and H), ARSER, JTK, and LS (C–E, I–K) are all run through Metacycle to produce an average period prediction meta2d (F and L). Data for *Arabidopsis* (blue) and wheat (yellow) was filtered for BH q < 0.01 on a data window of 24–68 h after dawn. (Data_Fig_S3a-l in S2 Data). (PDF)

**S4 Fig. Distributions of circadian phases of wheat transcript expression within period bins.** Rhythmic genes were defined as having Metacycle q-values of <0.01 over 24–68 h data relative to dawn. To improve accuracy, genes were placed into a period bin if both meta2d (Metacycle) and FFT-NLLS (Biodare2) predicted period lengths within the same 2 h window for each gene and were discarded if they fell into different bins. Meta2d phases were then recalculated relative to the meta2d period length (phase * 24/period) per gene. Histograms show that genes with shorter period lengths tended to have more dawn-peaking genes and genes with longer periods tended to have more dusk peaking genes. (Data_Fig_S4 in S2 Data). (PDF)

**S5 Fig. Comparison of balanced triads as classified by Ramírez-González and colleagues (2020) and circadian balanced triads classified in this study.** A chi-square test was used to determine whether there was a significant difference between the proportion of balanced and imbalanced triads (as defined by Ramírez-González (2020)) in each of the circadian-balanced and circadian-imbalanced triad categories. Results indicated that 83.20% of circadian-balanced triads were also categorised as balanced in the Ramírez-González data, whereas 64.15% of circadian-imbalanced triads were categorised as balanced in the Ramírez-González data. This difference was significant, $\chi 2(2) = 689.8$, $p < 0.00001$. These data also show that a large proportion of circadian triads labelled as balanced in the Ramírez-González dataset would be expected to have imbalanced patterns of expression if measured over several time points under circadian conditions, highlighting the importance of considering temporal dynamics in transcriptomic studies. Ramírez-González data is from Chinese spring leaves (excluding flag-leaf) under non-stressed conditions to provide as close a match to our data as possible. The diurnal time of collection is unknown. (Data_Fig_S5 in S2 Data). (PDF)

**S6 Fig. Examples of how definition of transcriptional triad balance can change in circadian time.** Each point represent expression normalised to 1 within each triad. Lines connect the dynamic changes in balance across the time course. Triad 59: A suppressed across all time points, but dominance of B and D vary across the time course. All homoeologs are classified as rhythmic (q < 0.05) but have imbalanced phases. Triad 176: Mostly appears as balanced across the time course but occasionally looks as though A is suppressed. All homeologs in this triad are rhythmic (q < 0.05) but have imbalanced relative amplitudes and periods. Triad 179: B suppressed over most time points, but when A homoeolog has a trough of expression appears as balanced. This triad has imbalanced phases and periods but all 3 homeologs are rhythmic (q < 0.05). Triad 10885: In this case, the A homoeolog is antiphase to the other homoeologs,

and so in dawn time points the triad appears balanced, but in dusk time points there is A dominance. Ternary plots were created using https://www.ternaryplot.com/. (Data_Fig_S6 in S2 Data).
(PDF)

**S7 Fig. Expression of homeologs of key transcription factors and their putative downstream targets.** Wheat WCBF2 (A) putatively regulates *WDHN17* (B) and *WRAB13* (C), TaPCF5 (D) putatively regulates wheat orthologs of *HY5* (E), *NIA2* (F), *NIA1* (G), and *XTH27* (H). (Data_Fig_S7 in S2 Data).
(PDF)

**S8 Fig. Loom plots showing positions of runs of imbalanced circadian triads.** Triads in the following categories are shown: 1A, 1B, 1D [triads with 1 rhythmic gene on the A, B, or D chromosomes, respectively and 2 arrhythmic genes], 2AB, 2AD, 2BD [triads with 2 rhythmic genes on the AB, AD, or BD chromosomes, respectively and 1 arrhythmic gene]. Points indicate that the homoeolog is rhythmic, and coloured lines represent the category of the triad.
(PDF)

**S9 Fig. *Arabidopsis* and wheat modules with a peak lag of 0 (synchronous phase) share more GO-slim terms than non-synchronous modules.** Numbers of significantly enriched ($p < 0.05$) GO-slim terms in common between 9 pairwise modules in *Arabidopsis* and wheat were counted and the pairwise modules were grouped based on the highest correlation score (peak lag) following cross-correlation with a lag of 0, 4, 8, or 12 h. (Data_Fig_S9 in S2 Data).
(PDF)

**S10 Fig. Transcription factor superfamily enrichment in wheat modules.** Barcharts showing number of genes (y axis) belonging to each TF superfamily (x axis) within wheat co-expression modules W1-W9. Coloured bars denote those superfamilies significantly over-enriched (red) or under-enriched (blue) compared to the total number of TF superfamilies present in the 16,327 genes submitted to WGCNA (Fisher's exact test, $p < = 0.05$). Families not significantly over- or under-enriched are coloured grey. (Data_Fig_S10 in S2 Data).
(PDF)

**S11 Fig. TFBS superfamily enrichment in wheat modules.** Bar charts showing number of nonredundant TFBS motifs (y axis) belonging to each TFBS superfamily (x axis) within wheat co-expression modules W1-W9. Coloured bars denote those superfamilies significantly over-enriched (red) or under-enriched (blue) compared to the total number of nonredundant TFBS motif superfamilies present in the 16,327 genes submitted to WGCNA (Fisher's exact test, $p < = 0.05$). Families not significantly over- or under-enriched are coloured grey. (Data_Fig_S11 in S2 Data).
(PDF)

**S12 Fig. Phylogenetic relationships for circadian genes based on alignment of LHY-like MYB full-length proteins.** The species identifiers for each species have been abbreviated as follows: Taestivum: *Triticum aestivum* (hexaploid wheat), Bdistachyon: *Brachypodium distachyon*, Hvulgare: *Hordeum vulgare* (barley), Osativa: *Oryza sativa* (rice), Stuberosum: *Solanum tuberosum* (potato), Athaliana: *Arabidopsis thaliana*. Wheat genes have been highlighted in blue and *Arabidopsis* genes in red for clarity. Bootstrap values are calculated using RAxML, with values over 50 shown on branches.
(PDF)

**S13 Fig. Phylogenetic relationships for circadian genes based on alignment of TOC1-like and PRR full-length proteins.** The species identifiers for each species have been abbreviated as follows: Taestivum: *Triticum aestivum* (hexaploid wheat), Bdistachyon: *Brachypodium distachyon*, Hvulgare: *Hordeum vulgare* (barley), Osativa: *Oryza sativa* (rice), Stuberosum: *Solanum tuberosum* (potato), Athaliana: *Arabidopsis thaliana*. Wheat genes have been highlighted in blue and *Arabidopsis* genes in red for clarity. Bootstrap values are calculated using RAxML, with values over 50 shown on branches.
(PDF)

**S14 Fig. Phylogenetic relationships for circadian genes based on alignment of PAS/LOV protein domains.** The species identifiers for each species have been abbreviated as follows: Taestivum: *Triticum aestivum* (hexaploid wheat), Bdistachyon: *Brachypodium distachyon*, Hvulgare: *Hordeum vulgare* (barley), Osativa: *Oryza sativa* (rice), Stuberosum: *Solanum tuberosum* (potato), Athaliana: *Arabidopsis thaliana*. Wheat genes have been highlighted in blue and *Arabidopsis* genes in red for clarity. Bootstrap values are calculated using RAxML, with values over 50 shown on branches.
(PDF)

**S15 Fig. Phylogenetic relationships for circadian genes based on alignment of LUX-like MYB full-length proteins.** The species identifiers for each species have been abbreviated as follows: Taestivum: *Triticum aestivum* (hexaploid wheat), Bdistachyon: *Brachypodium distachyon*, Hvulgare: *Hordeum vulgare* (barley), Osativa: *Oryza sativa* (rice), Stuberosum: *Solanum tuberosum* (potato), Athaliana: *Arabidopsis thaliana*. Wheat genes have been highlighted in blue and *Arabidopsis* genes in red for clarity. Bootstrap values are calculated using RAxML, with values over 50 shown on branches.
(PDF)

**S16 Fig. Phylogenetic relationships for circadian genes based on alignment of ELF4-like protein domains.** The species identifiers for each species have been abbreviated as follows: Taestivum: *Triticum aestivum* (hexaploid wheat), Bdistachyon: *Brachypodium distachyon*, Hvulgare: *Hordeum vulgare* (barley), Osativa: *Oryza sativa* (rice), Stuberosum: *Solanum tuberosum* (potato), Athaliana: *Arabidopsis thaliana*. Wheat genes have been highlighted in blue and *Arabidopsis* genes in red for clarity. Bootstrap values are calculated using RAxML, with values over 50 shown on branches.
(PDF)

**S17 Fig. Additional free-running expression of core circadian clock genes in wheat and their homologs in *Arabidopsis*.** Supplementary to Fig 3 in the main text. Wheat circadian clock genes were identified through alignment of phylogenetic protein family trees or BLASTP to known clock gene homologs. Wheat homoeologs are coloured according to their identity to either the A genome (orange), B genome (yellow), or D genome (blue), and grey and white blocks indicate subjective dark and light time periods under constant conditions. Data represent the mean of 3 biological replicates and transcript expression is collapsed to gene level. (Data_Fig_S17 in S2 Data).
(PDF)

**S18 Fig. *Arabidopsis* and wheat photosystem genes under free-running L:L conditions.** Data is mean normalised TPM. Shaded white and grey backgrounds indicate perceived day and night periods, respectively. Gene IDs for all genes plotted can be seen in S10 Table. (Data_Fig_S18-20 in S2 Data).
(PDF)

**S19 Fig.** *Arabidopsis* **and wheat light signalling genes under free-running L:L conditions.** Data is mean normalised TPM. Shaded white and grey backgrounds indicate perceived day and night periods, respectively. Gene IDs for all genes plotted can be seen in S10 Table. (Data_Fig_S18-20 in S2 Data).
(PDF)

**S20 Fig.** *Arabidopsis* **and wheat primary metabolism genes under free-running L:L conditions.** Data is mean normalised TPM. Shaded white and grey backgrounds indicate perceived day and night periods, respectively. Gene IDs for all genes plotted can be seen in S10 Table. (Data_Fig_S18-20 in S2 Data).
(PDF)

**S1 Table. Proportions of rhythmically classified genes using datasets filtered to remove low-expression genes.** Numbers of rhythmic genes at ($q < 0.05$ or $q < 0.01$) in *Arabidopsis* and wheat identified using Metacycle Benjamini–Hochberg q-values. Data windows reflect hours relative to transfer to constant light. Metacycle was used on expression datasets with genes removed that had fewer than 0.1 TPM in more than 6 time points. The effects of filtering data to remove low-expression data are discussed in S1 Note.
(XLSX)

**S2 Table. Meta2d estimates from previously published circadian datasets.** Estimates are based on gene-level TPM normalised transcripts from a data window of 24–68 h relative to transfer to constant light for each species. Rhythmic genes are defined as having a q-value < 0.01 and period and rAMP means are for rhythmic genes only.
(XLSX)

**S3 Table. GO-slim terms for biological processes associated with similar patterns of circadian expression or different patterns of circadian expression in *Brassica rapa* paralogs.** Only enriched terms which were highly enriched (Fisher's exact test $p < 0.01$) in 1 category and nonsignificantly expressed ($p > 0.05$) in the other category are displayed.
(XLSX)

**S4 Table. Expected and observed runs of rhythmic triad categories across the wheat genome.** Rhythmic imbalanced triads were classified as having 1 or 2 rhythmic homoeologs $q < 0.01$ where the other homoeolog has a $q > 0.05$. Triads with balanced rhythmicity have 3 rhythmic homoeologs $q < 0.05$. In total, chromosome position data was found for 4,225 of these triads. Expected maximum run lengths for each category were calculated as described in S5 Note. For Category test 3, Triads lacking rhythmicity on the A chromosome are either: triads with 1 rhythmic homeolog on the B subgenome, triads with 1 rhythmic homoeolog on the D subgenome or triads with 2 rhythmic homoeologs on the B and D subgenomes. The same logic applies to the other 2 categories. *P*-values for randomness in each category set were computed using a Wald–Wolfowitz test using publicly available code: https://github.com/psinger/RunsTest and implemented in Python.
(XLSX)

**S5 Table. Circadian characteristics of module eigengenes.** Expression data (TPM) over a time window of 24–68 h after transfer to constant light (L:L) was filtered for rhythmicity (BH $q < 0.01$) before clustering in WGCNA. Eigengene expression values were analysed using Metacycle to approximate phase and period values.
(XLSX)

**S6 Table. GO-slim terms for each of the 9 modules identified in *Arabidopsis* and wheat.**
(XLSX)

**S7 Table. Gene IDs for core circadian genes and corresponding *Arabidopsis* orthologs.**
(XLSX)

**S8 Table. Metacycle (Meta2d) estimates for core circadian genes and corresponding *Arabidopsis* orthologs.**
(XLSX)

**S9 Table. Enriched GO-slim terms in rhythmically expressed genes in *Arabidopsis* and wheat.**
(XLSX)

**S10 Table. Expression data and Metacycle statistics for transcripts involved in key circadian regulated pathways.**
(XLSX)

**S11 Table. Summary table with expression of wheat genes (TPM), Metacycle estimates across (0–68 h and 24–68 h), gene annotations, and triad balance classification https://opendata.earlham.ac.uk/opendata/data/wheat_circadian_Rees_2021.**
(XLSX)

**S12 Table. TPM matrix for all wheat genes at replicate level https://opendata.earlham.ac.uk/opendata/data/wheat_circadian_Rees_2021.**
(CSV)

**S1 Data. Data used in main figures in the manuscript including replicate level data, means, standard deviations, and mean normalisation where appropriate.**
(XLSX)

**S2 Data. Data used in Supplementary figures including replicate level data, means, standard deviations, and mean normalisation where appropriate.**
(XLSX)

## Acknowledgments

We thank Dr. Susan Duncan for her contribution to preliminary experiments prior to the start of this project.

## Author Contributions

**Conceptualization:** Hannah Rees, Anthony Hall.

**Data curation:** Rachel Rusholme-Pilcher, Connor Reynolds.

**Formal analysis:** Hannah Rees, Rachel Rusholme-Pilcher.

**Funding acquisition:** Anthony Hall.

**Investigation:** Hannah Rees, Rachel Rusholme-Pilcher, Paul Bailey, Calum A. Graham, Luíza Lane de Barros Dantas, Antony N. Dodd.

**Methodology:** Hannah Rees, Joshua Colmer, Sabrina Jaye Ward.

**Project administration:** Anthony Hall.

**Supervision:** Hannah Rees, Anthony Hall.

**Validation:** Joshua Colmer, Benjamen White, Benedict Coombes.

**Visualization:** Hannah Rees.

**Writing – original draft:** Hannah Rees.

**Writing – review & editing:** Hannah Rees, Luíza Lane de Barros Dantas, Antony N. Dodd, Anthony Hall.

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
