## [Editor Report · Decision Letter 0]

6 Jun 2022

Dear Dr Hall, 

Thank you for submitting your manuscript entitled "Circadian regulation of the transcriptome in a complex polyploid crop" for consideration as a Research Article by PLOS Biology, and apologies again for our delay in sending you an initial decision.

Your manuscript, reviews from Review Commons, and Response to Reviewers have now been evaluated by the PLOS Biology editorial staff. I am writing to let you know that we are, in principle, interested in the study and would therefore like to send your revised manuscript back to the reviewers from Review Commons.

Once your full submission is complete, your paper will undergo a series of checks in preparation for peer review. After your manuscript has passed the checks it will be sent out for review. To provide the metadata for your submission, please Login to Editorial Manager (https://www.editorialmanager.com/pbiology) within two working days, i.e. by Jun 08 2022 11:59PM.

**IMPORTANT**, please also address the following editorial requests: 

1) When completing your submission, please make sure to include a track-changes version of the manuscript, showing the changes made during the revision. This should be uploaded as a "Related" file type.

2) After some discussion with the team, we think your study may be best suited for our Methods and Resources article type (https://journals.plos.org/plosbiology/s/what-we-publish#loc-methods-and-resources-articles). If you agree, we ask that you change the article type accordingly.

Kind regards,

Lucas

Lucas Smith, Ph.D.

Associate Editor

PLOS Biology

lsmith@plos.org

---

## [Decision Letter · Decision Letter 1]

22 Jul 2022

Dear Dr Hall,

Thank you for your patience while we considered your revised manuscript "Circadian regulation of the transcriptome in a complex polyploid crop" for publication as a Methods and Resources at PLOS Biology. This revised version of your manuscript has been evaluated by the PLOS Biology editors, the Academic Editor and the original reviewers from Review Commons.

You can read the reviewer comments below my signature. As you will see, the reviewers feel the revision has largely addressed their original concerns and they highlight that this is a valuable dataset. However, the reviewers have identified a number of minor issues which we think should be addressed before publication, in a revision that we think will not take very long. 

Based on the reviews, we are likely to accept this manuscript for publication, provided you satisfactorily address the remaining points raised by the reviewers. 

**IMPORANT: In addition to addressing the reviewer comments please also make sure to address the following data request: 

--

DATA POLICY REQUEST:

Fig 1b,c,h-n; Fig 2b; Fig 3a-m; Fig4a-f; Fig S1a-c; Fig S2a-f; Fig 3a-l; Fig S4; Fig S5; Fig S6; Fig S7; Fig S9; Fig S10; Fig S11; Fig S17a-m; Fig S18a-d; Fig s19a-h; Fig S20a-n

>>Please also ensure that figure legends in your manuscript include information on where the underlying data can be found, and ensure your supplemental data file/s has a legend.

>>Please ensure that your Data Statement in the submission system accurately describes where your data can be found.

--

**We expect to receive your revised manuscript within two weeks. However, if you need additional time to address the reviewer concerns, please do let us know and we are happy to grant an extension.**

*Published Peer Review History*

*Press*

Sincerely,

Luke

Lucas Smith, Ph.D.

Associate Editor,

lsmith@plos.org,

PLOS Biology

Reviewer remarks:

Reviewer #1: I appreciate the work done by Rees et al. to address and respond to my concerns from the first review. This dataset will be a fantastic resource for the community and will hopefully inspire others to perform circadian studies in other crops as well as other cultivars within a species.

I have a few minor comments to improve clarity in the text:

1. In figure 1, panels l-m. It would be helpful to have the color legend directly on the figure to easily identify the subgenomes.

2. This is just a comment, but I love supplemental figure 6 and think this is really cool and a great way to emphasize the importance of time course data and being carefully about assessing subgenome dominance from single time point (or single condition) studies.

3. For Figure 3 and supplemental figure 17, it is difficult to make statements about expression level when there is no statistical support. Given that these are normalized plots without SD showing the variation I would suggest either running a test or de-emphasizing the expression level and focusing more on the period and phase. 

4. On line 383 referring to HFR1 expression the Figure reference should be 4e not 4c.

5. The results section on starch metabolism has some redundancies that could be eliminated with a little reorganizing. Currently, starch regulation is explained at the start of the section and then there is a shift to discussing Rubisco and then returning to starch synthases. It would improve flow to bring all the relevant starch metabolism text together.

Reviewer #2: The authors have addressed all of my comments and concerns sufficiently.

In one of the edits a parenthesis needs closing (line 146).

As stated in my first review of this paper on Review Commons I believe that this is a very valuable, and carefully constructed resource to the circadian biology community and beyond, including crop improvement.

Reviewer #3: After revision by the authors, I have no objections to the manuscript.

---

## [Editor Report · Decision Letter 2]

18 Aug 2022

Dear Dr Hall,

Thank you for the submission of your revised Methods and Resources article, "Circadian regulation of the transcriptome in a complex polyploid crop", for publication in PLOS Biology. On behalf of my colleagues and the Academic Editor, Pamela Ronald, I am pleased to say that we can in principle accept your manuscript for publication, provided you address any remaining formatting and reporting issues. These will be detailed in an email you should receive within 2-3 business days from our colleagues in the journal operations team; no action is required from you until then. Please note that we will not be able to formally accept your manuscript and schedule it for publication until you have completed any requested changes.

PRESS

Sincerely, 

Lucas Smith, Ph.D., Ph.D.

Associate Editor

PLOS Biology

lsmith@plos.org